

# Western boundary circulation and coastal sea-level variability in northern hemisphere oceans

Samuel Tiéfolo Diabaté[1], Didier Swingedouw[2], Joël Jean-Marie Hirschi[3], Aurélie Duchez[3], Philip J. Leadbitter[4], Ivan D. Haigh[5], and Gerard D. McCarthy[1]

[1]ICARUS, Department of Geography, Maynooth University, Maynooth, Co. Kildare, Ireland
[2]Environnements et Paleoenvironnements Oceaniques et Continentaux (EPOC), UMR CNRS 5805 EPOC-OASU-Universite de Bordeaux, Allée Geoffroy Saint-Hilaire, Pessac 33615, France
[3]National Oceanography Centre, Southampton, UK
[4]University of East Anglia, Norwich, UK
[5]University of Southampton, Southampton, UK

**Correspondence:** Samuel Tiéfolo Diabaté (samuel.diabate.2020@mumail.ie)

**Abstract.** The northwest basins of the Atlantic and Pacific oceans are regions of intense Western Boundary Currents (WBC), the Gulf Stream and the Kuroshio. The variability of these poleward currents and their extension in the open ocean is of major importance to the climate system. It is largely dominated by in-phase meridional shifts downstream of the points where they separate from the coast. Tide gauges on the adjacent coastlines have measured the inshore sea level for many decades and

provide a unique window on the past of the oceanic circulation. The relationship between coastal sea level and the variability of the western boundary currents has been previously studied in each basin separately but comparison between the two basins is missing. Here we show for each basin, that the inshore sea level upstream the separation points is in sustained agreement with the meridional shifts of the western boundary current extension over the period studied, *i.e.* the past seven (five) decades in the Atlantic (Pacific). Decomposition of the coastal sea level into principal components allows us to discriminate this variability

in the upstream sea level from other sources of variability such as the influence of large meanders in the Pacific. This result suggests that prediction of inshore sea-level changes could be improved by the inclusion of meridional shifts of the western boundary current extensions as predictors. Conversely, long duration tide gauges, such as Key West, Fernandina Beach or Hosojima could be used as proxies for the past meridional shifts of the western boundary current extensions.

## 1 Introduction

Western boundary currents (WBCs) are a major feature of global ocean circulation and play an important role in global climate by redistributing warm salty waters from the tropics to higher latitudes. The role of WBCs in the redistribution of heat and salt in the Atlantic is an integral part of the Atlantic Meridional Overturning Circulation (AMOC), resulting in heat transported towards the equator in the South Atlantic and the largest heat transport of any ocean northwards in the North Atlantic (Bryden and Imawaki, 2001). WBCs also interact strongly with the atmosphere, influencing regional and global climate variability

(Imawaki et al., 2013; Kwon et al., 2010; Czaja et al., 2019) and impact the sea level of the coastlines they are adjacent to (Little et al., 2019; Sasaki et al., 2014; Woodworth et al., 2019; Collins et al., 2019).





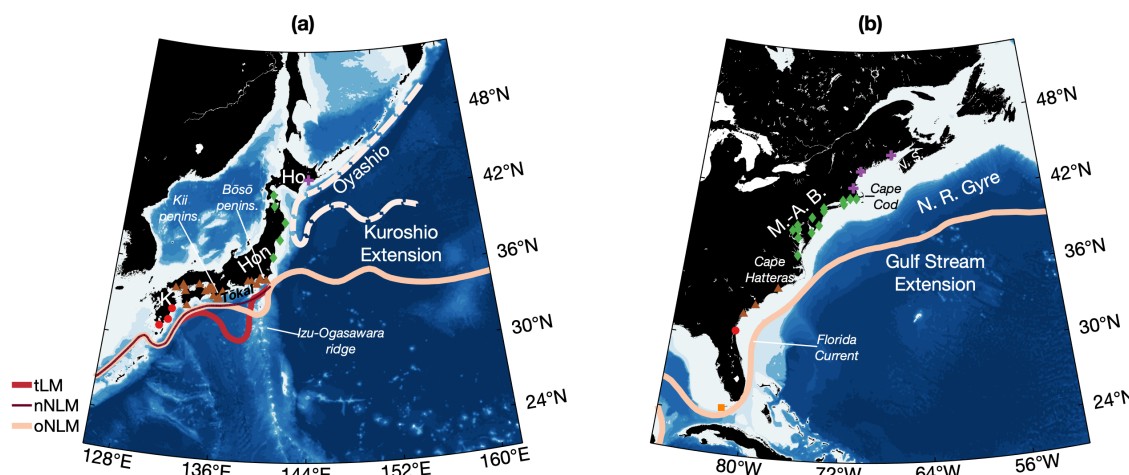

**Figure 1.** (a) Kuroshio region circulation: The three Kuroshio paths — the typical Large Meander (tLM), the near-shore Non-Large Meander (nNLM) and the offshore Non-Large Meander (oNLM) are indicated upstream of the Izu-Ogasawara Ridge. The mean location of the KE is indicated offshore of this point showing the location of the quasi-stationary meanders. The Oyashio current is shown in blue. On land, K indicates Kyūshū, Hon stands for Honshū and Ho indicates Hokkaidō. (b) Gulf Stream region circulation from the Florida Current to the Gulf Stream Extension. The Northern Recirculation Gyre is also indicated. On land, M.-A. B. stands for Mid-Atlantic Bight and N.S. indicates Nova Scotia. Markers in (a) and (b) indicate the location of the tide gauges used in this study. The colour and shape of the markers in (a) and (b) indicate the angle used to rotate the wind stress in an alongshore/across-shore coordinate system for the removal of sea-level variability driven by local atmospheric effect (See Supplementary Table S1 and Supplementary Table S2). Shadings in (a) and (b) indicate bathymetry.

In the Pacific, north of 30°N, the Kuroshio flows northeastwards along the coast of mainland Japan before leaving the coast at approximately 35°N and becoming a separated boundary current known as the Kuroshio Extension (KE, Figure 1 (a)). The Kuroshio and KE have variable flow regimes including decadal timescale variability, with the KE following either a stable and northern path, or an unstable and southern path (Qiu et al., 2014; Imawaki et al., 2013; Kawabe, 1985). This variability is driven by the wind stress curl over the central North Pacific which generates Sea Surface Height (SSH) anomalies. These anomalies progress westward as jet-trapped waves, shifting meridionally the KE before reaching the Kuroshio – Oyashio confluence (Sugimoto and Hanawa, 2009; Sasaki et al., 2013; Sasaki and Schneider, 2011a; Ceballos et al., 2009). Southeast of Japan, negative (positive) SSH anomalies ultimately displace the Kuroshio southward (northward) above the shallower (deeper) region of the Izu-Ogasawara Ridge (IOR). Interaction of the Kuroshio with the bathymetry when it is shifted above the shallower region of the IOR is possibly the cause of an unstable Kuroshio Extension (Sugimoto and Hanawa, 2012). In any case, when the KE is unstable, it has a more southern mean position, and the Kuroshio follows the offshore Non-Large Meander (oNLM) path (see Fig. 1 (a)). When unstable the Kuroshio has a lower overall transport (Sugimoto and Hanawa, 2012), which has an impact on the associated ocean heat transport. When the KE is stable, it exhibits a quasi-stationary meanders and a more northern mean position, and the Kuroshio south of Japan tends to follow either the typical Large Meander (tLM) or the near-shore Non-Large Meander (nNLM) (Sugimoto and Hanawa, 2012; Qiu et al., 2014; Usui et al., 2013).



Among the typical paths that the Kuroshio can take south of Japan (Fig. 1), the typical large meander is without doubt the most remarkable, and is a major driver of the regional sea level (Kawabe, 2005, 1995, 1985) and atmospheric variability (Sugimoto et al., 2019). Large meanders (LM) occur when two stationary eddies strengthen south of Japan. One is located
southeast of Kyūshū and associated with an anticyclonic circulation; and the other one is located south of Tōkai and associated with a cyclonic circulation. The front bounded by the two eddies becomes the Kuroshio large meander, and thus the cyclonic anomaly is inshore between the Kuroshio path and the southern coasts of Tōkai.

In the Atlantic, the Gulf Stream has its origins in the eponymous Gulf of Mexico, flowing past the Florida coastline as the Florida Current before leaving the boundary at Cape Hatteras near 35°N. From here it flows eastward as a meandering, eddying,
free current in the Gulf Stream Extension, and eventually the North Atlantic Current. From the American coast to 60°W – 55°W, northward or southward lateral motions of the Gulf Stream Extension dominate its interannual and seasonal variability. This notable intrinsic variability follows closely the main mode of Atlantic atmospheric variability: the North Atlantic Oscillation (NAO) (Joyce et al., 2000; McCarthy et al., 2018). The abrupt transition from warm subtropical waters to cold subpolar waters marks a 'North Wall' of the Gulf Stream (Fuglister, 1955). This Gulf Stream North Wall (GSNW) is a convenient marker of
the lateral motions of the Gulf Stream Extension (Frankignoul et al., 2001; Joyce et al., 2000; Sasaki and Schneider, 2011b). The horizontal circulation of separated western boundary current interact closely with the vertical circulation. The vertical circulation in this region is part of the AMOC which can be simplified as northward flowing Gulf Stream waters and southward flowing deep waters as part of the Deep Western Boundary Current (DWBC). One paradigm of the interaction of vertical and horizontal circulation in the region is that an enhanced DWBC, enhanced AMOC, 'pushes' the GSNW to the south, and
expands the Northern Recirculation Gyre (NRG). However, diverse behavior has been found in models, with some supporting this paradigm (Zhang and Vallis, 2007; Zhang, 2008; Sanchez-Franks and Zhang, 2015) and some finding the opposite: an enhanced AMOC, northward shifted GSNW (De Coetlogon et al., 2006; Kwon and Frankignoul, 2014). Alternatively, as in the Pacific with the KE, the Gulf Stream Extension has been linked to the mechanism of remote wind stress curl forcing the westward propagation of large-scale jet undulations (Sasaki and Schneider, 2011b). Finally, Andres et al. (2013) higlighted that
the coastal sea level on the large shelf north of Cape Hatteras was in agreement with the location of the Gulf Stream Extension west of 69 °W and suggested that the shelf transport 'pushes' the Gulf Stream, whereas Ezer et al. (2013) hypothesized that a more inertial Gulf Stream south of the separation point may 'overshoot' to the north when leaving the coastline at 35°N and control, at least to some extent, the location of the extension.

While the Gulf Stream and Kuroshio are western boundary currents driven by the closure of the Sverdrup balance (Stommel,
1948; Munk, 1950), even the brief introduction presented here highlights both differences and similarities between the currents. Upstream of separation point, the currents behave quite differently. The Kuroshio takes a number of distinct paths, whereas the Gulf Stream hugs the coast tightly. The separation point at the Bōsō peninsula and Cape Hatteras has a remarkably similar latitude both at 35°N. Downstream, the Gulf Stream Extension flows northeastward, whereas the Kuroshio Extension is mainly flowing eastward. The meandering of the Kuroshio in its extension region is much more defined than that of the Gulf Stream
Extension, with no named quasi-stationary meanders in the Gulf Stream Extension (until farther downstream at the Mann eddy). The north – south shifts of the extensions are remarkable features of both basins and account for an important part the





extensions' variability. It is well established that these lateral shifts are caused by the propagation of long jet-trapped waves forced by downstream wind in the Pacific, whereas the mechanisms driving the GSNW are not completely clear, with plausible role of a similar mechanism of wind-forced jet undulation. These jet-trapped waves are possible thanks to the sharp background

velocity gradient induced by WBC Extensions, comparable or greater to the meridional gradient of planetary vorticity within the mid-latitude band. Hence, the jet-trapped waves are essentially Rossby waves, but they propagate in the waveguide formed by the WBC extension, which allow their meridional narrowing as they progress westward, and their southwestward flow in the Atlantic (Sasaki et al., 2013; Sasaki and Schneider, 2011a, b). It is however important to note that, in the Atlantic, the lateral shifts of the Gulf Stream Extension have been more often linked with the DWBC and the NRG. In the Pacific, southern

(northern) shifts of the Kuroshio Extension are known to be concurrent with periods of instability (stability), whereas, until recent years (prior to ∼2000), the Gulf Stream Extension has been much more stable (Andres, 2016; Gangopadhyay et al., 2019). The interaction with the cold currents to the north is also quite different. The continent north of the Gulf Stream to Newfoundland lends to a topographical constraint on the gyre circulation, whereas the Oyashio is much less constrained by land. Conversely, the upstream Kuroshio is much more constrained than the upstream Gulf Stream, due to the presence of the

Izu-Ogasawara Ridge. Additionally, there is no Pacific equivalent to the coastal circulation on the prominent shelf north of Cape Hatteras (Peña-Molino and Joyce, 2008). The AMOC is a notably Atlantic-specific feature but there is not a distinct feature of the horizontal circulation that identifies clearly with the presence of the AMOC in the Atlantic basin that is not present in the Pacific basin. While a decline in the AMOC is robust in climate projections, WBCs are also expected to change. WBCs have been observed to be shifting polewards (Wu et al., 2012; Stocker et al., 2013) and becoming more unstable (Andres, 2016; Beal

and Elipot, 2016; Gangopadhyay et al., 2019).

Tide gauges estimate relative sea level at the coast and have done so since the 18[th] century in certain locations (*e.g.* Amsterdam, Stockholm, Kronstadt, Liverpool, Brest). Tide gauges have long been used to investigate ocean circulation in regions such as the Gulf Stream where the impact of strong ocean circulation on coastal sea level is apparent (Montgomery, 1938). However, ocean circulation is far from the only impact on sea level at the coast. The effects of land motion (including glacial isostatic

adjustment), thermosteric expansion, terrestrial freshwater changes (including river runoff and ice melt), and gravitational fingerprints all feature in sea level variations at the coast (Meyssignac et al., 2017). In addition, the local forcing of the atmosphere drives an important part of the coastal sea-level variability, particularly in shelf environments. Variations in wind stress can force water to travel toward (or away from) the coastline, consequently raising (lowering) the sea level at tide gauge locations. Both across-shore and alongshore wind stresses can impact sea level as can variations in the local air pressure through the

Inverse Barometer (IB) effect. On the American northeast coast, the inverted barometer greatly influences interannual change in the mean sea level, dominates most extreme interannual changes, and is not negligible on multidecadal timescales (Piecuch and Ponte, 2015), while the alongshore wind is also believed to play a role (Andres et al., 2013; Woodworth et al., 2014; Piecuch et al., 2019). This contribution of the atmosphere to the mean sea level is particularly challenging to disentangle from the contribution of ocean dynamics, because the two share similar range of timescales. Hence great care is needed to interpret

coastal sea level fluctuations, as measured by tide gauges, as representative of ocean circulation patterns.





A number of approaches have been developed to investigate ocean circulation using tide gauge data. The cross-stream gradient of sea level can be estimated by using an onshore tide gauge and an offshore island tide gauge (Montgomery, 1938; Kawabe, 1988; Ezer, 2015; Marsh et al., 2017) providing a direct estimate of a boundary current flowing between the gauges via the geostrophic relationship. This type of estimate is restricted to locations where suitable offshore island tide gauges exist.

Apart from the limited number of such locations, the offshore estimate is located in the eddy-filled ocean interior which can experience sea level fluctuations driven by the ocean mesoscale (Sturges and Hong, 1995; Firing et al., 2004) that are not representative of the large-scale ocean circulation. In the Atlantic, a number of studies (*e.g.* Bingham and Hughes, 2009; Ezer, 2013; McCarthy et al., 2015) have used long tide gauge records to estimate the strength of the AMOC, which has only been continuously observed since 2004 (Cunningham et al., 2007). In the Pacific, the difference between the sea level either side

of the Kii peninsula (Fig. 1) has been extensively used to diagnose past occurrence of the typical large meander (Moriyasu, 1958, 1961; Kawabe, 1985, 1995, 2005), despite the causal relationship not being fully understood.

Recent advances have been made on the theoretical underpinning of the relationship between sea level at the western boundaries of ocean and the offshore processes that influence sea level fluctuations (Minobe et al., 2017; Wise et al., 2018). The rule of thumb of Minobe et al. (2017) for a western boundary of the northern hemisphere is as follows: the sea level at a point on

the coastline is influenced by (1) long Rossby waves (or any other mass input from the east) incident on that point and (2) coastally trapped waves, transmitting equatorward the sea level signal from points farther to the north which, equally, can be influenced by incidental long Rossby waves. It follows that the alongshore gradient of the coastal sea level at a given latitude is proportional to the sea level input from the east at the same latitude (Minobe et al., 2017),

$$\frac{\partial}{\partial y}\left(\frac{\zeta}{f}\right)\bigg|_{x_W} = -\left(\frac{\beta}{f^2}\zeta\right)\bigg|_{x_I}, \qquad (1)$$

where $\zeta$ is the sea-level anomaly, evaluated at the coast ($x_W$) and at the frontier between the boundary layer and the ocean interior ($x_I$), and $\beta$ is the meridional gradient of the Coriolis frequency $f$. In the real ocean, the mass input into the western boundary region is more accurately described by the jet-trapped Rossby wave framework than by the direct westward propagation of linear long Rossby waves (Sasaki et al., 2013; Sasaki and Schneider, 2011a; Taguchi et al., 2007). Therefore, pairing the jet-trapped theory with Minobe et al. (2017) framework is expected to better estimate the sea level on the coast of western

boundaries. In accordance with this idea, the coastal sea level south of Japan is known to be in agreement with the Kuroshio location above the Izu-Ogasawara Ridge (Kuroda et al., 2010), the KE meridional shifts during the satellite era (Sasaki et al., 2014) and the regime shifts in North Pacific mid-latitude (Senjyu et al., 1999). Simply put, the mechanism is that jet-trapped long waves, originating from the east and responsible for the meridional shifts of the WBC extension, break, when reaching the coastline, in coastally trapped waves that propagate equatorward (Sasaki et al., 2014).

Globally, the mean sea level has shown an increased rate of rise in the last decades (Dangendorf et al., 2019; Nerem et al., 2018) induced by anthropogenic emission of greenhouse gases in the atmosphere, which is a major issue for coastal communities and environments. Understanding the relationship between sea level and ocean circulation is a component of understanding coastal vulnerability to changing sea levels. Many densely populated regions border WBCs and large changes in WBCs





could have big sea-level impacts. In the northern hemisphere, the Gulf Stream and Kuroshio border the US and Japan eastern
seaboards, two of the most densely populated coastlines in the world.

In this study, we analyse datasets of mean sea level along US and Japanese eastern coastlines, identify major spatial modes
of variability, and interpret this in terms of ocean circulation variability. This paper is organized as follows. Section 2 presents
the data used in this study and the derivation of indices for the WBC extensions. The results of the analysis of the gauge
records and their relationship to upstream and downstream WBC variability are presented and discussed in Sect. 3 and Sect. 4.
A conclusion is presented in Sect. 5.

## 2 Data and Methods

### 2.1 Tide gauge selection, treatment, and adjustment for surge variability

Tide gauge data were obtained from the Permanent Service for Mean Sea Level (Holgate et al., 2013, PSMSL, 2020, https:
//www.psmsl.org/) on the 17[th] August 2020. We selected tide gauge stations along the western boundary of the North Atlantic,
on the coast of the United States and Canada; and along the western boundary of the North Pacific, on the coast of Japan.
To retain only measurements of sufficient quality, length and completeness, historical series with more than 10% of missing
monthly values as well as those flagged for quality issues are excluded. Consequently, the number of individual tide gauge
records available is dependent of the chosen period. Summary of gauge details are given in Supplementary Table S1 and
Supplementary Table S2.

For the Atlantic, the period considered is January 1948 – December 2019 due to the availability of the atmospheric reanalysis
used to correct for surge effects. The island station on Key West is located onshore of the Gulf Stream and features a signal
coherent with the Florida tide gauges. It is therefore included, leaving a total number of 22 stations on the American east coast.

The Japanese tide gauge network is more recent, therefore the period considered for the Pacific is January 1968 – December
2019. The records of most stations on the eastern coast of Honshū feature important offset shift and/or drift after the March
2011 tsunami and cannot be used as such, leaving a ~700 km coastline strip depleted of any measurement. To remedy the
issue, we retained the three long records of Onahama, Miyako II and Ayukawa and replaced existing or missing data after
February 2011 with the closest SSH measurement, with trend and offset adjusted. The total number of tide gauges retained for
the Pacific region is 30 after the criterion of completeness is applied.

Missing values are linearly interpolated for both regions. No adjustment for long-term processes affecting the sea level
is performed as the records are quadratically detrended. Monthly anomalies are obtained by subtracting the climatological
monthly mean.

To correct the records from the effect of local winds and pressure, monthly sea-level pressure and ten meters above sea
level wind speeds were obtained from the NCEP/NCAR Reanalysis 1 (Kalnay et al., 1996, NOAA/OAR/ESRL PSL, https:
//psl.noaa.gov/). They are available from 1948 to present-day. The grid has a resolution of 1.875° in longitude and ~1.904° in
latitude. The variables are detrended and deseasonalized, after the wind speed is converted to wind stress. The Japanese 55-
year Reanalysis (Kobayashi et al., 2015, JRA-55, https://jra.kishou.go.jp/index.html), available from 1958, was also retrieved.



Results were similar to those obtained with the NCEP variables and are therefore not discussed. To assess and remove the local atmospheric contribution to changes in sea level, each monthly sea-level record is regressed against the atmospheric pressure and the wind stress interpolated at the gauge location, following the method of Dangendorf et al. (2013, 2014), Frederikse et al. (2017) and Piecuch et al. (2019). Details are given in the appendix (Sect. A), together with a brief analysis of the results. We find that the local forcing of the atmosphere drives between 30% and 50% of the monthly sea-level variability at tide gauges located north of Cape Hatteras, the separation point of the Gulf Stream, and at tide gauges located north of 38°N on the eastern coast of Japan, whereas the atmospheric influence is reduced upstream of the separation points of the Kuroshio and Gulf Stream. The unexplained residual represents the sea level corrected for local atmospheric effects.

Finally, to focus on inter-annual and slower variations, a low-pass 19-months Tukey filter is applied. The filtering effectively reduces the period analysed to November 1968 – January 2019 for the Japanese gauges, and November 1948 – January 2019 for the American gauges.

## 2.2 Additional datasets

Gridded monthly Sea Surface Height (SSH), Temperature (SST) and Velocities (SSV) derived from satellite altimetry are available from 1993 and were obtained from the Copernicus Marine Environment Monitoring Service (CMEMS) website (https://marine.copernicus.eu). SSH and SST are obtained from the ARMOR3D product (Guinehut et al., 2012). SSV is retrieved from the REP L4 product and consists of the sum of geostrophic and modelled-derived Ekman components (Rio et al., 2014). The three datasets have a regular grid of 0.25° × 0.25°.

To examine the Gulf Stream and Kuroshio Extensions variability, the EN4 quality controlled subsurface (200m depth) ocean temperature profiles (Good et al., 2013, EN4.2.1) are used. The 1981 – 2010 objectively analysed mean temperature at 200 m depth field from the World Ocean Atlas 2018 (Locarnini et al., 2018, WOA18) is used to derive the climatological position of the Gulf Stream and Kuroshio Extensions between, respectively, 75°W and 55°W, and 141°E and 161°E. The climatological extension positions are defined as the whole number isotherm roughly corresponding to the surface velocity maximum, that is 17°C in the Atlantic and 16°C in the Pacific. The two datasets were downloaded on the 8[th] September 2020. Derivation of jet latitudinal position indices is detailed in Sect. 2.3

Finally, we retrieved additional indices that infer oceanic and atmospheric variability. We make use of the GSNW index from Joyce et al. (2000) and of the Kuroshio Extension indices from Qiu et al. (2016), and also derive in Sect. 2.3 indices for the variability of the two WBC extensions. Additionally, the estimate of the southernmost latitude of the Kuroshio axis south of Tōkai (136°E – 140°E) produced by the Japan Meteorological Agency (JMA) was retrieved from their website (https://www.data.jma.go.jp/gmd/kaiyou/data/shindan/b_2/kuroshio_stream/kuro_slat.txt, the website was last accessed in January 2020). This index represents the upstream meridional movements of the Kuroshio south of Japan. We also retrieved the monthly North Atlantic Oscillation (NAO) index from James Hurrell and National Center for Atmospheric Research Staff (Eds) NAO webpage (https://climatedataguide.ucar.edu/climate-data/hurrell-north-atlantic-oscillation-nao-index-station-based).

As the tide gauge records and the reanalysis variables, satellite observations are deseasonalized, detrended and filtered with a 19-months Tukey filter, unless otherwise mentioned. All aforementioned indices are similarly deseasonalized, detrended and





filtered, with a couple of exceptions: yearly indices cannot be filtered at such a high cut-off frequency, and the indices of Qiu et al. (2016) feature no seasonal variability and are therefore not deseasonalized.

The significance of correlations between two timeseries A and B is calculated using the non-parametric method of Ebisuzaki (1997), as was previously done in McCarthy et al. (2015). The method consists of evaluating the Fourier transform of A
and generating a large number (here 5000 is used) of random timeseries with similar spectral properties. The modulus is preserved while the phase is randomized. The randomly generated signals are then correlated against B. Significance for zero-lag correlation between A and B is given as the percentage of randomly generated correlations which are less than the correlation between A and B (using absolute values). When we report lagged correlations, we use a more stringent test of confidence, as McCarthy et al. (2015). In this case, the lead-lag correlation between each randomly generated signal and B
is computed, and the maximum correlation is determined. The significance is given as the percentage of randomly generated maximum correlations which are inferior to the maximum correlation between $A$ and $B$ (within the limit of a lead, or lag, of a fourth of A or B length).

### 2.3 Meridional motions of the Western Boundary Current Extensions

At interannual to multidecadal scale, the Gulf Stream Extension and the Kuroshio Extension are quite similarly characterized
by strong lateral movements. The displacements are of about half a degree in the Atlantic (Joyce et al., 2000) and about one degree in the Pacific (Sasaki et al., 2013), with an increase in the meridional extent of the shifts toward the east.

For each ocean, the methods used to quantify such oscillations have evolved differently. In the North Atlantic, the Gulf Stream North Wall (GSNW) is defined as the leading mode of the temperature anomaly at the climatological position of the jet, or, more traditionally, its northern front (the 'North Wall'). Indeed, because the WBC extensions separate cold water to the
north from warm water to the south, warming (cooling) at the climatological jet position reflects a northern (southern) shift of the jet. In the Pacific, recent work has used SSH estimates averaged in the 31°N – 36°N and 140°E – 165°E box as proxy to infer the past Kuroshio Extension meridional location (Qiu et al., 2014, 2016). This area corresponds to the Kuroshio Southern Recirculation Gyre (KSRG) of which the strength is a good indicator not only of the Kuroshio Extension latitudinal location, but also of its stability and intensity (Qiu et al., 2014).
To produce consistent indices for both oceans, we made use of the subsurface sparse temperature observations to derive up-to-date indices of the meridional location of the Kuroshio Extension and Gulf Stream Extension, following the GSNW calculation method of Sasaki and Schneider (2011b) and Frankignoul et al. (2001). Given the data availability, the analysis period was restricted to 1960 and 1965 onwards for the Atlantic and Pacific respectively. For each year up to 2019, the available sparse subsurface temperature observations were interpolated at the climatological position of the Gulf Stream and Kuroshio
Extensions using an inverse distance weighting technique with power parameter $p = 2$ and a search radius of 400 km, allowing construction of an along-jet temperature matrix. The search radius acts as a spatial low-pass filter and was purposely set well above the Rossby deformation radius, to minimize the meso-scale meandering variability in the gridded temperature anomaly. The leading mode of variability is extracted for each basin by performing an Empirical Orthogonal Function (EOF) decomposition based on correlation (rather than covariance) on the detrended temperature anomaly. Figure 2 present the associated





EOFs (a and b) and principal components (c and d, light blue lines), for the Atlantic and the Pacific respectively. The EOF amplitude in both oceans varies in-phase all along the climatological jet axis. In the remainder of this paper, we refer to the principal components as GSNW and KE Index (KEI), and specify "this study/our GSNW" or "this study/our KEI" whenever precision is needed.

To contextualize the temporal variations of our GSNW and KEI with existing indices, we retrieved the GSNW estimate
of Joyce et al. (2000), available from 1955 to 2011, as well as the two KEIs of Qiu et al. (2016). The KEIs of Qiu et al. (2016) are derived from the KSRG strength as above-mentioned and were initially introduced by Qiu et al. (2014) using satellite altimetry and model output. They have been recently made available from 1905 to 2015 (1945 to 2012) using wind (Temperature/Salinity) data (Qiu et al., 2016). The wind-based index is obtained by forcing a 1½-layer reduced-gravity model with historical wind stress merged from ERA-20C and Interim reanalysis sets. Although such model has limited skills in
reproducing the westward narrowing of the meridional jet oscillations (see Sasaki et al., 2013; Sasaki and Schneider, 2011a), it is able to correctly reproduce the timing of the meridional shift of the KE (Taguchi et al., 2007). The three indices are presented alongside the GSNW (this study) and KEI (this study) in Figure 2 (c) and (d), after detrending is applied. They are yearly averaged for comparison with our GSNW and KEI. Correlations between both Qiu et al. (2016) indices and our KEI are high, with greater value obtained with the T/S index, (r = 0.75, significance is above 99%), than with the wind-based index
(r = 0.67, significance is 99%). Similar correlation is found between Joyce et al. (2000) GSNW and this study GSNW over their overlapping period, 0.71 (significance >99%).

## 3  Results

In this section, we propose a scrutiny of the inshore sea level measured by tide gauges using cross-correlation and moving correlation analysis, as well as Empirical Orthogonal Function (EOF) analysis. We relate the obtained spatial and temporal
patterns to ocean circulation. Senjyu et al. (1999); Valle-Levinson et al. (2017) and Sasaki et al. (2014) used EOF decomposition on the Pacific and the Atlantic tide gauges, and hence our analysis can be seen as building on their work.

### 3.1  Cross-correlation analysis

Correlation analysis has been performed extensively for tide gauges on the US East Coast from Woodworth et al. (2014) to McCarthy et al. (2015), Piecuch et al. (2016), and Calafat et al. (2018), among others. The resultant correlation patterns suggest
groupings of tide gauges across geographic regions, with boundaries defined by changing oceanographic circulation regimes, which we argue is the fingerprint of ocean circulation on coastal sea level.

We expand the analysis to tide gauges along the Japanese Coast. Three tide gauge groupings are apparent on Figure 3 (a), based on the cross-correlation between Japanese records. West of the Kii peninsula, the correlation between gauges is on average 0.85. From the Kii peninsula to the Bōsō peninsula, region that we refer to as Tōkai for simplicity, another highly
correlated group exists. The mean of the correlations within that group is 0.74, with the tide gauge of Owase showing slightly lower agreement. These two groups south of Japan were identified by the early work of Moriyasu (1958). The gauges on





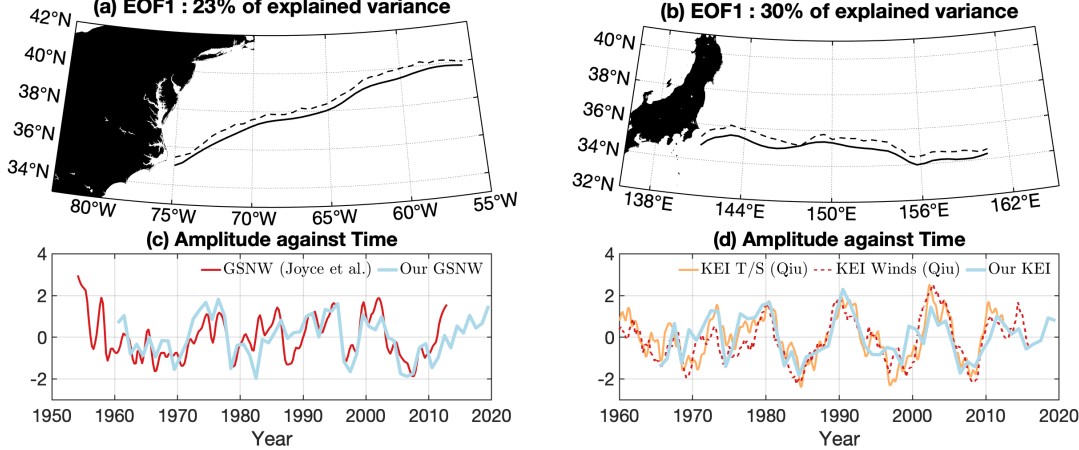

**Figure 2.** (a) GS climatological position (solid line) and GS climatological position plus GSNW spatial amplitude based on EOF analysis of the along-jet temperature anomalies (dashed line). Conversion ratio from normalized temperature to latitude is arbitrarily set for visualisation, but is the same at every grid point. (c) GSNW time series obtained from EOF analysis (thick light blue), with also the GSNW index of Joyce et al. (2000) shown in red. (b) and (d) As for (a) and (c), but obtained for the Kuroshio Extension, and with the KE indices of Qiu et al. (2016) replacing Joyce et al. (2000). The solid orange line in (d) corresponds to the temperature and salinity based index, and the dashed thin red line corresponds to the wind-based index. All timeseries in (c) and (d) are normalized.

the eastern coastline of Honshū and Hokkaidō, including the tide gauge of Katsuura, east of the Bōsō peninsula, show lower correlation with each other and are referred to as the Oyashio group. The limits of the correlation groupings match two oceanographic boundaries, the Kii peninsula, where the Kuroshio detaches during large meander periods, and the region of the Bōsō
peninsula, where the Kuroshio leaves the coast to become the Kuroshio Extension (see Fig. 1).

In the Atlantic, the pattern of correlation previously highlighted by McCarthy et al. (2015) and Woodworth et al. (2014) of a drop in correlation between tide gauges north and south of Cape Hatteras — the point where the Gulf Stream leaves the coastline — is also seen in our analysis, with a distinct change in correlation patterns noted either side of Cape Hatteras (Figure 3 (b)). All gauges south of Cape Hatteras display almost identical behaviour with correlation average within that group
equal to 0.78. All the gauges north of Cape Hatteras display high correlations as well, on average 0.72. The correlation of the three gauges located north of Cape Cod with the others north of Cape Hatteras is lower (on average r = 0.64 including the tide gauge of Boston and r = 0.58 without). This indicates that another boundary exist at Cape Cod, although the transition is not as abrupt as across Cape Hatteras.

In the Atlantic, Thompson and Mitchum (2014), Frankcombe and Dijkstra (2009), as well as Häkkinen (2000) noted high
correlation of sea-level variations from Nova Scotia (Canada) to the Caribbean. This, of course, implies correlation across Cape Hatteras. While we do find a few significant correlation at the 95% level across Cape Hatteras, indicated by bold outlines in Figure 3, the distinct drop in correlation is more prominent. We therefore investigate the evolution of the correlation patterns

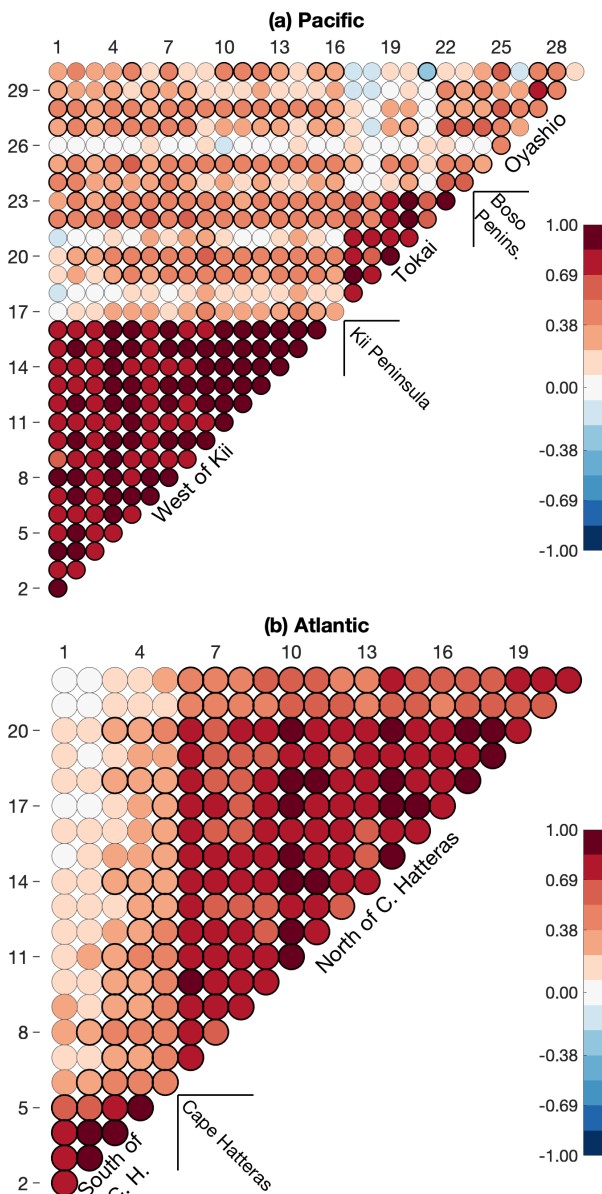

**Figure 3.** (a) Pacific and (b) Atlantic tide gauge cross-correlation at zero-lag. Each circular marker gives the the correlation $r_{ij}$, between gauge $i$ and gauge $j$. Tide gauge are numbered in ascending order from south to north, following the coastline. Bolder circular contour indicates that the correlation is above significance level of 95%.

through time in Figure 4 and Supplementary Figure S1. Within the groups (a) south and (b) north of Cape Hatteras, the individual correlations (Supplementary Figure S1 (a) and (b), thin grey lines) are high and show little time dependency, with the median never dropping below 0.54 south of Cape Hatteras and below 0.65 north (solid red lines). Figure 4 (b) shows the



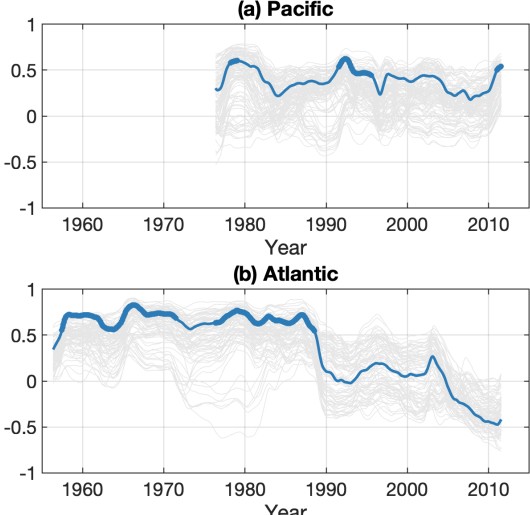

**Figure 4.** Moving correlation analysis between the tide gauges records with a running window of 15 years. Each individual grey line renders the moving correlation between two records. The x-axis represents the center of the moving window. (a) Cross correlation between the Oyashio and West of Kii groupings (grey lines). (b) Cross-correlation between gauges on either side of the Cape Hatteras, the separation point of the Gulf Stream (again, grey lines) On each panel, the solid blue line is the moving correlation between the group averages of sea-level anomalies. Significant correlations above 95% level are indicated by thicker blue line.

changing correlations between gauges located on either side of Cape Hatteras (thin grey line), as well as the moving correlation between the two grouping averages. The correlations are high and the moving correlation between the two grouping averages is most of the time significant from roughly the 1950s to the late 1980s, where the correlations drop abruptly. From the 1990s onwards, there is no correlation between both regions, with the exception of the most recent era, from approximately 2010, when the correlation is negative. This change in the behaviour was also noted by Kenigson et al. (2018) (see their supplementary material).

In the Pacific, the correlations within the two southern groupings feature little time variations (Supplementary Figure S1 (c) and (d)). The large deviations between individual correlations within the Oyashio group underline the overall lower agreement between the gauges there (panel (e)). As will be discussed further below, the sea level south of Tōkai is largely affected by the appearance of large meanders, which is a phenomenon unique to the Pacific. Thus, to compare southern and northern variability as was done for the Atlantic in Figure 4 (b), we plotted the changing correlations between the group west of Kii and the Oyashio group in Figure 4 (a). The correlations are relatively low over the whole period, with the correlation median exceeding r = 0.35 only at three occasions, 1978 – 1982, 1991 – 1993 and 2011 – 2012 (not shown), and the moving correlation between the two grouping averages rarely significant except in the early 1990s. More importantly, the moving correlations do not show an abrupt change, in contrast with the situation in the Atlantic.



## 3.2 Empirical orthogonal function analysis

We employ Empirical Orthogonal Function (EOF) analysis to objectively reduce the sea-level anomalies in an ensemble of modes, each composed of a time-varying coefficient $\alpha$, the *Principal Component* (PC), and associated spatial-varying coefficients $\phi$, the *Empirical Orthogonal Vector* or *Function* (EOF). Covariance-based EOF decomposition is performed on tide
gauge sea-level anomalies interpolated on a regular grid. This prevents the sea-level variability in better sampled region from being favored in the analysis. Details of the interpolation on a regular grid, including handling of estuarine stations, are given in Appendix B. The regular grid points are referred to as virtual stations.

The EOF analysis is computed separately on both Atlantic and Pacific gridded sea-level anomalies. Together, the two leading modes explain 85% (77%) of the variability of the Atlantic (Pacific) dataset.

### 3.2.1 Atlantic and Pacific first modes

The leading mode of the Pacific dataset explains 47% of the overall variance. The associated EOF, $\phi_1$, features a common region of coherence west of the Bōsō peninsula, south of Japan, where the Kuroshio separates from the coast (Figure 5 (a), circular markers). The Atlantic leading mode explains 60% of the variance and, in a similar way, $\phi_1$ features greater amplitudes south of the separation point, Cape Hatteras, and decreasing northward from there (Fig. 5 (b)).

On the southern coastline of Japan (130°E – 141°E), the amplitude is on average 2.3 cm (excluding the easternmost virtual station, located on the Bōsō peninsula $\phi_1 = 0.7$ cm). Towards the north, east of Honshū and Hokkaidō, the mode amplitude is reduced by a factor of four and equals 0.7 cm on average (this time including the virtual station on the Bōsō peninsula). The mode temporal variations $\alpha_1$ are presented on Figure 5 (c). There is a decrease from the late 1970s to 1985, followed by an increase until 1990. From there onwards, the principal component exhibits marked fluctuations with a $4 - 7$ year period. In the
Atlantic, the transition from the south of Cape Hatteras, where $\phi_1$ is on average 2.6 cm, to the north, on average 1.3 cm, is not as pronounced as with the Pacific gauges. The associated time-varying amplitude $\alpha_1$ was maximum in the mid-1970s, the mid-1980s and the early 1990s, and was particularly low in the mid 1960s, the early 1980s and during the 2000s (Fig. 5 (d)).

To demonstrate the link between the gauge records and the ocean dynamics, we computed two composites using the monthly surface velocity magnitude since the beginning of that record in 1993 (Rio et al., 2014). The surface velocity magnitudes
are averaged over periods of strongly positive $\alpha_1$ (greater than two third its standard deviation, *i.e.*, $\alpha_2 > 2/3$) to form a first composite, and similar procedure is done over periods of strongly negative $\alpha_1$ (lower than minus two third its standard deviation, *i.e.*, $\alpha_1 < -2/3$). The treshold of $\pm 2/3$ is arbitrary, but taking any other tresholds within $0 - 1$ lead to similar patterns. Colour shadings on Figure 5 (a – b) represent, for each basin, the difference between sea surface velocity composites based on each mode temporal amplitude.
Similar patterns are seen in both basins downstream of the separation point. In the Pacific, the positive velocities, east of the ridge (>140°E), are located farther north than the negative velocities, indicating that the Kuroshio Extension is found more to the north during period of positive $\alpha_1$. The surface velocity composite in the Atlantic presented alongside the EOF on Figure 5 (b) highlights an analogous situation, with the Gulf Stream Extension drifting to the north (positive velocity pattern) during





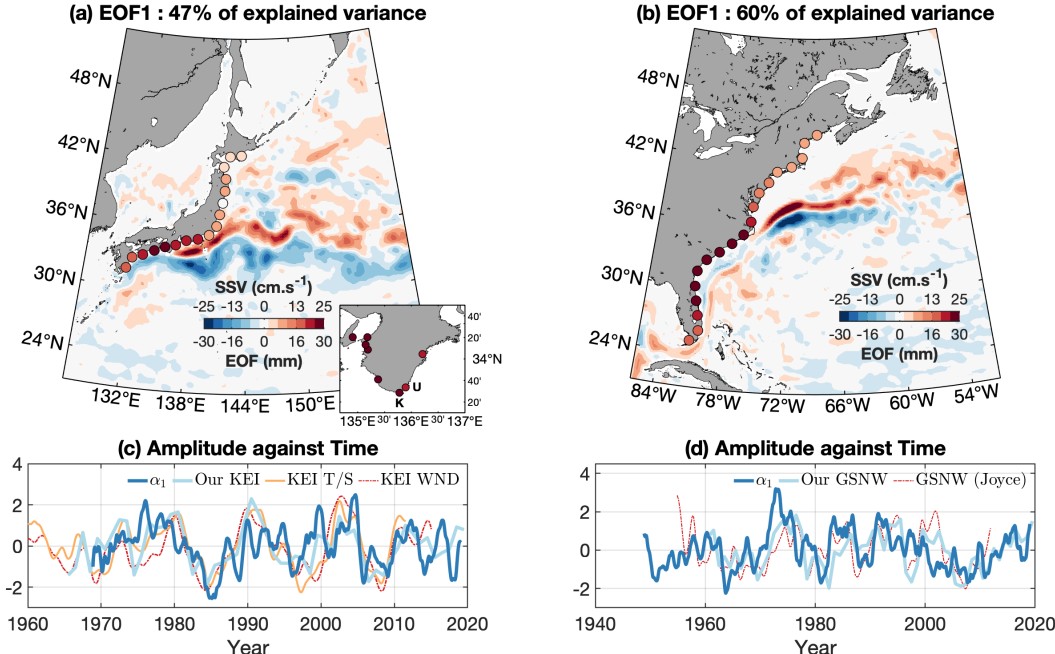

**Figure 5.** (a) and (b) present the leading EOF ($\phi_1$) of the gridded, tide gauge obtained, sea-level anomaly (circular markers). (c) and (d) present, for the Pacific and Atlantic respectively, the associated principal components $\alpha_1$'s (solid blue lines), together with indices of extension meridional location: this study and Qiu et al. (2016) T/S-based and wind-based KEIs (on (c), thick light blue line, orange solid line and dot-dashed red line respectively); and this study and Joyce et al. (2000) GSNWs (on (d), thick light blue line and dot-dashed red line, respectively) . All quantities on (c) and (d) are normalized. Colour shadings on (a) and (b) are, for each basin, the sea surface velocity magnitude composites difference based on the principal component $\alpha_1$ (period of $\alpha_1 > +2/3$ minus period of $\alpha_1 < -2/3$). The arbitrary threshold of $\pm 2/3$ was used, but taking any number between zero and one leads to akin patterns. The inset on (a) present the regression coefficient obtained when the principal component is regressed on the original tide gauges, with a zoom in on the Kii peninsula.

periods of strong positive $\alpha_1$ (early 1990s, mid 2010s) and to the south (negative velocity pattern) during periods of strong
negative $\alpha_1$ (2000s).

Different patterns are found upstream of the separation point. South of Japan and east of 136°E, a region of positive velocity exists close to the coast, to the north of negative velocities. In particular, in the Izu-Ogasawara Ridge region (140°E), positive velocities are above the deep channel located north of 34°N (deeper than 1500m) whereas negative velocities are spread above the shallower part of the ridge to the south (shallower than 1500m). The Kuroshio was hence found northward (southward)
during periods of positive (negative) $\alpha_1$. Moreover, it is obvious that the positive velocity pattern (associated with high $\alpha_1$) resembles the nearshore NLM (see Figure 1), whereas the negative velocity pattern (associated with low $\alpha_1$) resembles the offshore NLM. In the Atlantic, the negative velocity pattern is inshore of the positive velocity pattern upstream of Cape Hatteras, indicating that during periods of positive (negative) $\alpha_1$, the upstream Gulf Stream was offshore (inshore).





To extend prior to the satellite era the analysis of the relationship between the two principal components and the extension location, we make use of the Kuroshio Extension indices and Gulf Stream North Wall indices (this study; Qiu et al., 2016; Joyce et al., 2000) described in Sect. 2. The principal components are correlated against the indices, after they were yearly or quarterly averaged (when necessary). There is moderate but significant correlation between the Pacific $\alpha_1$ and the various KE indices. The maximum correlation with our KEI (Figure 5 (c), solid green curve) is r = 0.52 (significance is 99%) and is found at zero lag. Similarly, we obtain moderate correlation between $\alpha_1$ and the two Qiu et al. (2016) indices. There is better agreement with the index based on temperature and salinity (Figure 9 (c), orange line), with r = 0.52 when $\alpha_1$ leads by one month (significance is 99%, note that r = 0.52 for leads between zero and two months, with similar significance), than with the wind-based index (dot-dashed red line), for which the correlation maximum is found when $\alpha_1$ lags by 7 months and is r = 0.41 (61%). The correlation is not stationary through time however. The 1990s are a period of lesser agreement, as the sea-level features a bump which is not seen in KEIs. In contrast, the signals co-vary from 2000 onwards. Likewise, they all feature the strong shift of the late 1970s to mid-1980s.

Correspondingly, the Atlantic principal component $\alpha_1$ is in agreement with our GSNW index at no lag, r = 0.46 (99%), although greater correlation is obtained when $\alpha_1$ leads by one year (r = 0.52, significance is 99%). The agreement with Joyce et al. (2000) GSNW is more tenuous, with r = 0.30 at no lag (90%). The correlation maximum is 0.34 (68%) and is found when $\alpha_1$ leads by two years, although correlations above 0.27 are obtained in a lag range of minus three to plus one year (negative sign indicates $\alpha_1$ leads). The values obtained with Joyce et al. (2000) GSNW are not significant above 95% level, but we argue that the agreement at low-frequency is significant. For example, we took up again the EOF analysis, after the 19-months filter applied to the sea-level anomalies was substituted by a 73 months (∼6 years) filter. The obtained EOF $\phi_1$ is not greatly changed (Suppl. Fig. S2 (a)). The correlation between the obtained $\alpha_1$ and Joyce et al. (2000) (our) GSNW, similarly filtered, equals 0.61 (0.78) when $\alpha_1$ leads by one and a quarter of a year (one year) which is significant at 92% (99%).

We can conclude that the link between coastal sea-level upstream the separation point and the latitude of the jet extension downstream, which was highlighted in the Pacific by Sasaki et al. (2014) and Kuroda et al. (2010) is actually a feature of both basins, and extend before the satellite era.

### 3.2.2 Atlantic and Pacific second modes

While similar patterns emerge in both the Atlantic and Pacific leading modes, the same is not true for the second modes. The second EOFs $\phi_2$'s of the Pacific and Atlantic tide gauges are presented on Figure 6 (a) and (b) respectively. The EOF of the Pacific dataset is dominated by the tide gauges on the shores of the Tōkai district. This pattern corresponds to the group of correlation from Uragami to Yokosuka presented on Figure 3 (a). The averaged amplitude of the mode in the region is 2.5 cm, but rises to 3.0 cm when only the two virtual stations east of the Kii peninsula are considered. This indicates that, in the region South of Tōkai, the second mode is larger in magnitude to the leading mode. The second EOF of the Atlantic is dominated by the tide gauges north of Cape Hatteras, with amplitude of 1.7 cm on average north of 35°N. There is little deviations from the value of 1.7 cm, but, because the leading mode diminishes northward (Figure 5 (b)), the second mode dominates north of Cape Cod, whereas the two modes have similar magnitude in the Mid-Atlantic Bight.



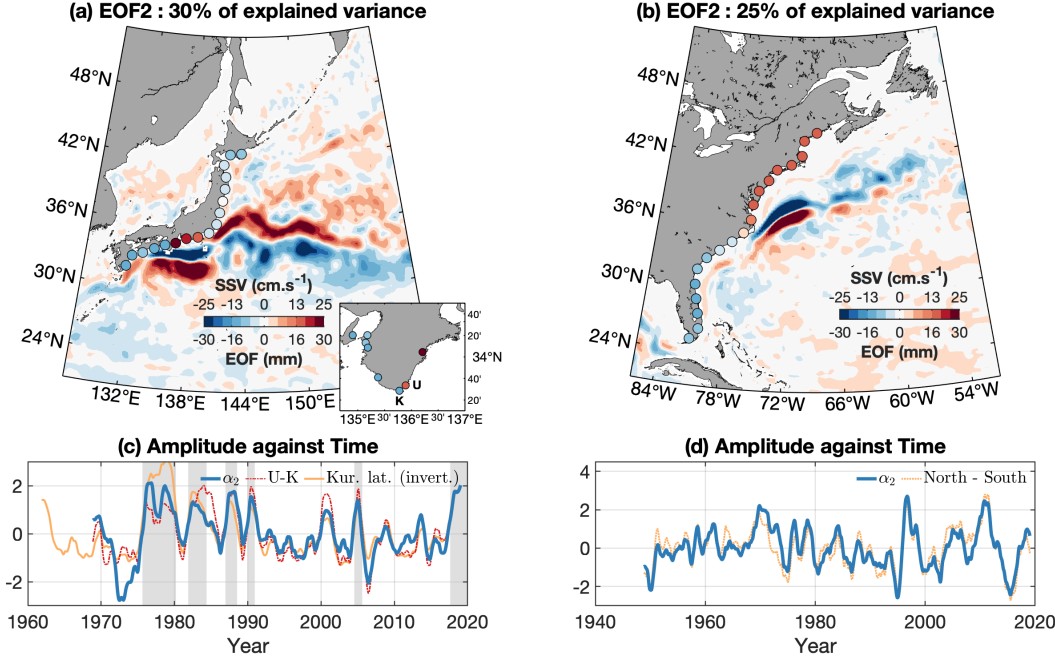

**Figure 6.** (a) and (b) present the second EOF ($\phi_2$) of the gridded, tide gauge obtained, sea-level anomaly (circular markers). (c) and (d) present, for the Pacific and Atlantic respectively, the associated principal components $\alpha_2$'s (solid blue lines). Also on (c), the Kuroshio southernmost latitude South of Tōkai is represented with the solid orange line (positive values indicate Kuroshio further to the south), the difference between sea level at Uragami and Kushimoto with the dot-dashed red line, and grey shadings represent periods of typical large meander (JMA, 2018). On (d), the dashed yellow line is the sea-level difference between the averages of northern and southern groupings of tide gauges, as in McCarthy et al. (2015). All timeseries on (c) and (d) are normalized. Colour shadings on (a) and (b) are, for each basin, the sea surface velocity magnitude composite difference based on the principal component $\alpha_2$ (period of $\alpha_2 > +2/3$ minus period of $\alpha_2 < -2/3$). The arbitrary threshold of $\pm 2/3$ was used, but taking any number between zero and one leads to akin patterns. The inset on (a) present the regression coefficient obtained when the principal component is regressed on the original tide gauges, with a zoom in on the Kii peninsula.

In the Pacific, a second group west of Kii varies in antiphase with the Tōkai gauges, with amplitudes on average of $-1.2$ cm. In the Atlantic, amplitudes south of Cape Hatteras are on average $-0.8$ cm (Figure 6 (a)). In both cases, the magnitudes of 385 these negative variations are more than 2 times smaller than to the magnitude of the positive variations (north of Cape Hatteras and south of Tōkai).

In the Pacific, large amplitude in the EOF $\phi_2$ are confined upstream the Kuroshio separation point, whereas the Atlantic mode is dominated by the variability passed the separation point, to the north. Furthermore, the velocity composite difference based on $\alpha_2$ and obtained in similar manner to the procedure aforementioned, present very different patterns from one basin to 390 another (colour shadings, Figure 6 (a) and (b)), which we return to in more detail below. As the two modes are different, we discuss them separately.



### 3.2.3 The second mode in the Pacific

The principal component $\alpha_2$ obtained with the Pacific gauges is closely linked with the typical large meander of the Kuroshio. Negative values coincide with known periods when the Kuroshio took the tLM pathway (as acknowledged by the Japanese

Meteorological Agency (JMA, 2018): August 1975 to March 1980, November 1981 to May 1984, December 1986 to July 1988, December 1989 to December 1990, July 2004 to August 2005, August 2017–2020, see shading on Figure 6 (c)) whereas the Kuroshio took the rest of the time one of the NLM pathways, or less often, an atypical path.

The principal component is extremely close (r = 0.83, significance well above 99%) to the difference in sea level between Kushimoto and Uragami (thin dot-dashed red line on Figure 6 (c)), two stations located either side of Cape Shiono-Misaki on

the Kii peninsula, which is the point of separation between the two groups of coherent variability highlighted in the previous section. The relationship between the tLM periods and the sea-level difference between those two stations is known since the early work of Moriyasu (1958, 1961) and was investigated by Kawabe (1985, 1995, 2005), among others. The inset on Figure 6 (a) presents the regression coefficient obtained when the principal component is regressed on the original tide gauge records, with a zoom in on the Kii peninsula. Despite geographical proximity (less than 20 km), the stations of Kushimoto and

Uragami (indicated by a K and a U on the inset) are affected very differently by the second mode. The amplitude at Uragami is negative and is positive at Kushimoto. On the other hand, as was discussed previously, the leading EOF is of same sign and relatively similar magnitude on all of the southern coast of Japan (see the inset of Figure 5 (a) for the amplitude of the leading mode at Kushimoto and Uragami), and the other modes have negligible amplitudes in the region. From there, subtracting the Kushimoto timeseries to the ones from Uragami essentially gets rid of the influence of the leading EOF and reveals the

underlying variability south of Tōkai.

The Japan Meteorological Agency estimate of the southernmost latitude of the Kuroshio axis south of Tōkai (136°E – 140°E) is shown on Figure 6 (c) (orange line, axis is inverted so that southern shift is a positive anomaly). Correlation with the principal component $\alpha_2$ is strong and highly significant ($r = 0.82$, significance >99%), confirming that the mode is a footprint of the large meander. Note that the correlation is slightly higher with the principal component than with the difference between

Kushimoto and Uragami ($r = 0.78$, significance >99%).

The velocity composite, derived on high (>2/3) minus low (<−2/3) values of $\alpha_2$ as was done for the leading modes, is shown in Figure 6 (a). When the principal component is strongly positive, *i.e* when the Tōkai coastal sea level is high, the Kuroshio south of Tōkai (135°E – 141°E) is found farther south than when the principal component is negative where it is found much closer to the coast. The positive velocity patch intersects the negative velocity patch, so that the Kuroshio path

east of the Bōsō Peninsula is closer to the coast during the positive $\alpha_2$ periods, and flows northeastward, whereas during the negative period, the Kuroshio flows essentially eastward. Simply put, the negative velocity pattern represents the NLMs and the positive velocity pattern, the LM (see Fig 1. The situation east of the ridge (>140°E) is roughly similar to Figure 5 (a) ; that is, the Kuroshio Extension, is found more to the north when the principal component is positive. The negative velocities are also more scattered than their positive counterparts, highlighting that the KE was more stable during period of positive $\alpha_2$

(see also Sugimoto and Hanawa, 2012).





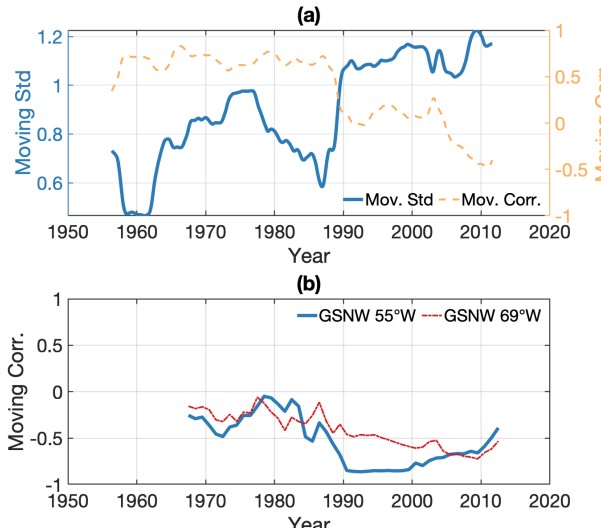

**Figure 7.** (a) Moving standard deviation of $\alpha_2$ with a 15 year running window (solid blue line) and moving correlation with the same windowing between the northern and southern Atlantic gauge grouping averages (red dashed line). The latter is the same as the blue line on Figure 4 (a). (b) Moving correlation with a 15 year running window between the second principal component $\alpha_2$ and our GSNW index (solid blue line), and between $\alpha_2$ and our GSNW index computed on 75°W – 69°W (dot-dashed red line).

### 3.2.4 The second mode in the Atlantic

The principal component associated with the second EOF in the Atlantic increases from 1948 to the early 1970s, followed by a decrease until the mid-1990s, with interannual deviations from those long-term changes (Figure 6 (d)). The mid-1990s mark an abrupt change, with the interannual variability increasing greatly in amplitude from then onwards. This is shown on Figure 7,

which presents the moving standard deviation of $\alpha_2$ obtained with a 15 year running window (solid blue line).

As noted by Valle-Levinson et al. (2017), the variability of this mode has already been shown in the past by the difference in the sea level either side of Cape Hatteras (McCarthy et al., 2015; Woodworth et al., 2017). The yellow dashed line on Fig. 6 (d) shows the difference between the averages of northern and southern gauge groupings. In comparison with the mean sea-level difference indices of McCarthy et al. (2015) and Woodworth et al. (2017) the timeseries are flipped, as we computed the

difference as north minus south, rather than the other way around. The agreement between the sea-level difference and $\alpha_2$ is very high, r = 0.88 (significant above 99%). As for the difference between Kushimoto and Uragami, substracting the sea level south of Cape Hatteras from north of Cape Hatteras (or reversely) minimizes the influence of the leading mode. Indeed, the difference between $\phi_1$'s magnitude either side of Cape Hatteras is 1.3 cm on average, whereas the difference between the EOF $\phi_2$ magnitude either side of Cape Hatteras is 2.5 cm.

The velocity composite exhibits two well-defined patterns of opposite sign off Cape Hatteras, indicating that the positive (negative) phases of $\alpha_2$ are concurrent with a southern (northern) shift of the Gulf Stream west of 69° W (Fig. 6 (b)). The





patterns bear some resemblance with the ones obtained with $\alpha_1$ and presented on Figure 5 (b), but the amplitudes of the composite along the Gulf Stream Extension make a strong contrast. The positive and negative velocity patches are now maximum in the region west of 69° W, where they have greater across-shore width than the ones obtained with $\alpha_1$ (Figure 5 (b)). There,

because the $-25$ to $25$ cm/s colour scale was retained for comparison with the other modes, the composite amplitudes are largely clipped. They are in the range of $(\pm)\,40-60$ cm/s, larger than the magnitudes obtained with $\alpha_1$ in the same region, which were in the range of $(\pm)\,10-40$ cm/s. On the other hand, composite amplitudes east of 69°W are smaller when obtained with $\alpha_2$ than when obtained with $\alpha_1$. In this second region, the $\alpha_2$-based composite is also less consistent, with the negative velocities intruding southward and splitting the positive pattern in two at $\sim$68°W. Hence, the link between $\alpha_2$ and the Gulf

Stream Extension meridional shifts in this second region is not as clear as the one obtained with $\alpha_1$.

## 4 Discussion

EOF analysis showed similar features of the leading mode of the two basins. The leading mode explained 47% of the variance in the Pacific gauges and 60% of the variability in the Atlantic gauges. Their spatial patterns were similar, having a greater amplitude south of the separation points than to the north (Figure 5). In both basins, the temporal amplitudes of these similar

modes were shown to co-vary with the meridional shifts of the western boundary current extension.

An important dissimilarity, however, is that the amplitude of the leading EOF in the Atlantic decreases gently north of the separation point, while the transition is abrupt in the Pacific. This result contrasts with the findings of Valle-Levinson et al. (2017), who obtained a leading EOF of the Atlantic gauge records with less marked northward decrease in amplitude. Although the different starting and ending periods may play a role, we find that the dissimilarity mostly arises because of

the correction we applied for surge-driven sea-level change (Appendix A). This result however, should not be interpreted as a demonstration that the atmosphere plays a role in extending the southern variability northward. Rather, the surge correction reduces the variance north of Cape Hatteras, which better constrains the EOF analysis and reduces undesired compensation between modes.

When the EOF analysis is recalculated restricting the period to $1960-1990$, where greater coherence either side of Cape

Hatteras is seen (Figure 4 (a)), the northward decrease in $\phi_1$ is still apparent. This is an important result, because previous studies had excluded the Gulf Stream and its extension as plausible drivers of the sea level on the western coast of the North Atlantic basin, on the basis that such drivers were not able to explain coherence across Cape Hatteras (Thompson and Mitchum, 2014; Valle-Levinson et al., 2017). On the contrary, we found that the Gulf Stream separation marks the point from where the mode's imprints of sea level diminishes, which re-qualifies the Gulf Stream presence as a plausible sea-level driver. In our

view, the northward decrease of $\phi_1$ is related to the orientation of the Gulf Stream Extension which gradually moves away from the shoreline north of 35°N. Following the same idea, the difference between the abrupt decrease of the EOF amplitude in the Pacific and the gradual decrease in the Atlantic arise from the different WBC Extension orientations, the Gulf Stream Extension flowing northeastward and the Kuroshio Extension, eastward.





The leading mode temporal amplitudes in both basin are in agreement with the location of the WBC extensions in both

altimetry-derived sea surface velocities and in-situ subsurface temperature. In the Pacific, anomalous wind stress curl triggers westward propagating baroclinic jet-trapped Rossby waves that shift the jet meridionally (Sasaki et al., 2013; Sugimoto and Hanawa, 2009; Ceballos et al., 2009). A similar mechanism has been proposed for the Atlantic (Sasaki and Schneider, 2011b). Sasaki et al. (2014) hypothesized that the incoming jet-trapped Rossby waves, which are responsible for the extensions' shifts, break on the western boundary and propagate equatorwards as Kelvin or other coastally trapped waves, linking the extension

variability to coastal sea level. Because the long jet-trapped Rossby waves provide a mass input to the western boundary which has a narrower meridional extent than traditional Rossby waves, the alongshore coastal sea level gradient is maximum near the WBC separation point (Equation 1). This leads to a 'shadow' coastal area north of the separation point which is less affected by the incoming jet-trapped wave, and an 'active' area which sees the progression of the coastal wave (Sasaki et al., 2014), explaining the EOF patterns of Figure 5. Hence, although Sasaki et al. (2014) focused on the KE and south Japan sea level,

our results support that such a mechanism could explain the link between the coastal sea-level and the extension meridional location observed in both oceans. It is true that we found that, in the Atlantic, the correlation between the GSNW and the leading principal component of the sea-level variability is maximum when the GSNW lags by about one year, but we must emphasize that we found significant correlation between the GSNW and the Atlantic $\alpha_1$ at zero-lag, in agreement with a mechanism of coastal wave following the jet-undulation.

The state-of-the-art theory of Sasaki et al. (2014) is elegant, but there is some limitations that we believe are useful to point out for future developments on the relationship between WBC extensions and upstream sea levels. First, the upstream bottom temperatures at the shelf break and on the shelf are known to covary with the upstream sea level (Kuroda et al., 2010). This limits the spectrum of possible coastally trapped waves to waves with non-zero cross-shore flow (*e.g.* topographic Rossby waves) which are able to drive warm water on and off the shelf. Secondly, the role played by the path variability upstream the

separation point is unclear. In the Pacific, the upstream patterns of velocity in Figure 5 (a) feature the offshore and nearshore NLM paths (Figure 1), which have not previously been understood as the propagation of SSH anomalies as coastally trapped waves. Furthermore, positive (negative) $\alpha_1$ values were shown to be concurrent to inshore (offshore) paths south of Japan, whereas south of Cape Hatteras, the opposite was observed (positive $\alpha_1$ associated with an offshore path). If coastally trapped waves indeed drive the upstream coastal sea level, it is conceivable that they also cause the concurrent offshore (inshore) shifts

of the western boundary currents upstream of their separation point. Yet, it is unclear why they would drive opposite behaviour in term of paths in the two basins, hence weakening the hypothesis. These issues underscore the difficulties to understand the causal relationship between the WBC extension and the upstream sea level.

The EOF analysis highlighted very different second modes in the two basins. The second EOF explained 30% of the variance in the Pacific gauges and 25% of the variability in the Atlantic gauges. In the Pacific, this second mode is the manifestation of

the meandering of the Kuroshio upstream of its separation point, whereas the second EOF in the Atlantic is mainly associated with variability north of Cape Hatteras, the separation point.

In the Pacific, the typical large meander influence on the sea level south of Tōkai was shown in our analysis, and is known since many decades (Moriyasu, 1958, 1961; Kawabe, 1985, 1995, 2005). The elevated values of $\alpha_2$ closely match the typical





large meander periods (Fig. 6 (c)), at the exception of April 2000 – April 2001 when $\alpha_2$ was high despite the Kuroshio not being
in a tLM phase. Sugimoto et al. (2019) highlighted that, during tLM phases, the strengthening of the anticyclonic circulation
accelerates a westward coastal current in 137°E – 140°E region, which allows the intrusion of warm Kuroshio water south
of Tōkai. Between 2000 and 2001, this westward flow, which closes the anticyclonic circulation south of Tōkai, can also be
observed. It is clearly apparent in monthly snapshots (Suppl. Fig. S3), and in the mean velocity between April 2000 and April
2001, although to a lesser extent (Suppl. Fig. S4 (b)). SST and SSH averages over this period show that the intrusion of warm
water south of Tōkai goes along with a rise of the SSH there, as do composites obtained over tLM periods (Suppl. Fig. S4 (c –
f)). In fact, the major distinction from tLM period is that between 2000 and 2001, the Kuroshio veered northward east of the
ridge (or on the ridge at 140°E). These types of pathway are sometimes called straddled large meander, because, in contrast
with typical large meanders, the anticyclonic eddy straddles the Izu-Ogasawara Ridge at 140° E.

The common denominator to the tLM and such atypical paths is the presence of the westward flow identified by Sugimoto
et al. (2019), which brings warm waters south of Tōkai. We hypothesize that the sea-level rise recorded by the gauges in the
region is forced by the intrusion of the Kuroshio warm water brought by such current (geostrophic tilting and/or steric rise).
From a coastal sea-level perspective, there is no qualitative difference between the forcing of typical large meander and the
forcing of atypical paths that straddle the ridge.

The second mode of variability of the Atlantic tide gauge is perhaps the most puzzling mode among the four discussed here.
As was indicated by Valle-Levinson et al. (2017), differentiating north minus south of Cape Hatteras sea level approximates this
mode well (McCarthy et al., 2015; Woodworth et al., 2017). However, the EOF $\phi_2$ has much greater absolute magnitude north
of Cape Hatteras than south, in contrast with Valle-Levinson et al. (2017). This indicates that simultaneous antivariations south
of Cape Hatteras are weak. Again, this disparity arises because of the correction we applied to remove local atmospherically
driven variability. It suggests that, at first order and for the period 1948 – 2019, differentiating the sea level across Cape Hatteras
removes the influence of the leading mode on the sea-level variability north of Cape Hatteras. In this region, $\phi_1$ and $\phi_2$ have
comparable amplitudes, despite the northward decrease in strength of the leading mode.

Our results indicate a clear change in the variance of $\alpha_2$ occurring around ∼1990 (Figure 7). Previous studies showed an
increase in the agreement between the North Atlantic Oscillation and the sea level variations north of Cape Hatteras occurring
around the same time (Andres et al., 2013; Kenigson et al., 2018). Studies that focused on the low frequency (7-year filtered)
mean sea-level difference across Cape Hatteras (McCarthy et al., 2015; Woodworth et al., 2017) noted a greater agreement with
the NAO since the second half of the twentieth century than before. For example, Woodworth et al. (2017) report a correlation
of −0.62 for the period 1950 – 2014 between the difference New York minus Key West and the NAO, and a lower correlation
of 0.21 for the period 1913 – 1949. It is reasonable to believe that the great agreement since ∼1990 contributes to the overall
low-frequency agreement since 1950, in conjunction with anticorrelated multidecadal variability: sea-level rise north of Cape
Hatteras (NAO decline) between 1948 and ∼1970, as well as between ∼1990 and ∼2010, and sea-level drop (NAO increase)
between ∼1970 and ∼1990.

Our analysis based on sea surface velocity composites highlighted the agreement of $\alpha_2$ with the Gulf Stream Extension
meridional location west of 69°W, consistently with Andres et al. (2013). The relevance of the satellite measurements for





interpretation of ocean dynamics prior to ∼1990 is however questionable. The sharp increase in the variance of the mode
around ∼1990 arises the issue whether this mode represents the pursuance of the same physical phenomenon throughout
the whole period 1948 – 2019, or if a mechanism supplanted another around ∼1990. We find that $\alpha_2$ has a non-stationary
relationship with this study GSNW index (Figure 7 (b), solid blue line). The correlation between the GSNW and $\alpha_2$ is of
−0.45 (significant above 99%) over the full period 1948 – 2019, quite similar to the correlation between the GSNW and $\alpha_1$,
but this agreement is due to the period after ∼1990, where correlation is r = −0.63 (>99%) while the correlation between
1948 and 1989 is −0.17 (61%). Hence, the relationship with the location of the Gulf Stream is largely limited to the recent
era, which complicates understanding of the forcing prior to $\alpha_2$ variance change around 1990. Note that here we use 1990 as
change point for simplicity, but similar results are obtained when using 1987 (Kenigson et al., 2018; Boon, 2012) or 1994,
which corresponds to the first strong negative $\alpha_2$ dip after more than 40 years.

One mechanism in particular has been hypothesized to tie the Gulf Stream location west of 69°W and the Nova Scotia to
Cape Hatteras sea level together since ∼1990: Andres et al. (2013) argued that the coastal sea level is proportional to the
geostrophic southward shelf transport, which interacts with the Gulf Stream at the separation point, where the shelf transport
and the Gulf Stream meet. The use of the inshore sea level alone to diagnose the shelf transport is supported by a variance
minimum in SSH anomaly located on the shelf break. Whether the Gulf Stream dictates the shelf sea level — and, hence, the
shelf southward transport — or the other way around is an open debate (Andres et al., 2013; Ezer et al., 2013; Peña-Molino
and Joyce, 2008).

Andres et al. (2013) hypothesized that the shelf transport is triggered by the alongshore wind forcing over the shelf, and
eventually drives the movements of the Gulf Stream to the south, rather than the opposite. A strong negative correlation be-
tween the coastal sea level north of Cape Hatteras and the alongshore wind stress over the northern part of the shelf supported
the hypothesis. Kenigson et al. (2018) highlighted that the year 1987 marked an abrupt change of the wind orientation above the
US northeast coast and Canadian east coast. If this hypothesis is indeed correct, it is intriguing that such sea-level variability
appears in our analysis, given that the tide gauge records have been corrected for instantaneous wind forcing, especially as
the sea-level response to the atmospheric forcing that was removed from the record (Appendix A) is quite different from $\alpha_2$
(Supplementary Figure S8 (b)). To investigate the question, we repeated the procedure of Andres et al. (2013) and correlated
the principal component against the detrended NCEP wind stress fields projected onto a 20° from zonal angle, roughly corre-
sponding to the orientation of the shelf. A 19-months filter was applied to the wind stress for compatibility with the principal
component. Supplementary Figure S5 (a) presents the correlation over the full period 1948 – 2019 between $\alpha_2$ and the along-
20° wind stress. The correlation above the shelf does not exceed 0.23. When the correlation is computed reducing the period to
1990 onwards, the patterns are greatly changed (Supplementary Figure S5 (c)), as noted by Andres et al. (2013) and Kenigson
et al. (2018). In contrast with Andres et al. (2013) however, the negative correlation pattern on Supplementary Figure S5 (c) is
575   shifted towards the Gulf of Saint Lawrence and east of Newfoundland, so that the agreement with the alongshore wind on the
shelf remains negligible everywhere. Hence, the variability of $\alpha_2$ is not due to the alongshore wind stress above the shelf, and
we can say that the latter has been successfully removed from the tide gauge records by our correction for local atmospheric
forcing. Furthermore, the role of remote wind stress in the Gulf of Saint Lawrence or east of Newfoundland is uncertain as





well, because at these latitudes the NAO is a confounding variable. When corrected for the NAO, the correlation between the
alongshore wind stress and $\alpha_2$ is not significant anywhere on sea (Figure Supplementary S5 (d)). This does not necessarily
exclude alongshore wind stress in the Gulf of Saint Lawrence or east of Newfoundland as a possible driver of the variability of
$\alpha_2$ since 1990, but indicates that any other forcing strongly correlated with the NAO is an as likely driver.

Alternatively, density anomalies formed in the Labrador Sea propagating southward along the western boundary, either as
coastally trapped waves or advected within the deep western boundary current, have been proposed as driver of the sea-level
variability north of Cape Hatteras (Frederikse et al., 2017). However, existing indices of the AMOC and DWBC do not show
the same variability as $\alpha_2$ (Caesar et al., 2018; Thornalley et al., 2018). Finally, we have not considered the role of salt (or lack
of) and water-volume input to the shelf caused by both river discharges and eddies detaching from the Gulf Stream, which are
an additional potential driver of the sea-level on the shelf (Piecuch et al., 2018).

EOF analysis can help to understand the major distinctions observed in the alongshore sea-level coherence between the two
basins. Upstream of the separation point in the Pacific, the cross-correlation analysis highlighted two distinct groupings either
side of the Kii peninsula. This is the point at which the Kuroshio path either follows the large meander or stays close to the
coast on a non-large meander path (Fig. 1). The EOF analysis revealed that the sea level in the region east of Kii is the sum of
the two leading mode, whereas in the region west of Kii the sea level is well approximated by the first mode only, where the
first mode is associated with the Kuroshio Extension meridional motions, and the following mode with meandering south of
Japan. Hence the second grouping of co-variability south of Japan is due to the emergence of (a)typical large meanders, which
are an additional forcing for the sea level east of Kii. This forcing has no equivalent in the Atlantic, and therefore there is only
one distinct grouping of variability south of Cape Hatteras.

In the Atlantic, the moving correlation analysis showed that the agreement between gauges south and north of Cape Hatteras
changed strongly around ∼1990. This is an additional distinction between the Atlantic and Pacific, as no coherence change of
such magnitude was observed between the Oyashio and West of Kii groupings in the Pacific. Figure 7 (a) presents the moving
correlation between the southern and northern gauge averages (dashed orange line) alongside the standard deviation of the
principal component $\alpha_2$ computed with a moving 15 years window (solid blue line). It is apparent that the change in variance
of $\alpha_2$ is concurrent with the shift in correlation between north and south of Cape Hatteras seen in the moving correlation
analysis. Hence, the change of coherence across Cape Hatteras around 1990 is not linked to a collapse of the leading mode, but
to an increase in the variability north of Cape Hatteras appearing in the second mode.

## 5 Conclusion

This study presents a consistent analysis of the two western boundary regions of northern Atlantic and northern Pacific. The
agreement between the upstream sea level and the WBC extensions' meridional shifts was highlighted in the two basins
conjointly. This agreement supports the mechanism of Sasaki et al. (2014) of trapped Rossby waves propagating within the
western boundary current extensions, shifting them meridionally en route and progressing into coastally trapped waves at arrival
at the coast, consequently modifying the inshore sea level. On the other hand, recent studies have linked the inshore upstream





sea level with other plausible drivers, including the ocean temperature within the western boundary current vicinity (Kuroda et al., 2010; Domingues et al., 2018) and the subtropical gyre interior sea surface height (Woodworth et al., 2014; Thompson and Mitchum, 2014; Volkov et al., 2019). It would be interesting to know what are the statistical and causal relationships

between these proposed drivers of the sea level and the shifts of the extensions of the western boundary currents. In the absence of such information, the mechanism proposed by Sasaki et al. (2014) is, so far, the only linking the upstream sea-level and the WBC extensions' meridional shifts. While this hypothesis is an important conceptual development, quantitative studies are for now missing. Hence, further work is required on the matter.

We showed that dissimilarities between Japanese and American inshore sea level emerge in the second mode of variability.

In the Pacific this relates to upstream meso-scale dynamics (Kuroshio large meander), whereas in the Atlantic, the second mode is mainly associated with changes north of Cape Hatteras, the separation point of the Gulf Stream, although weak antivariations exist to the south. In the Pacific and in comparison with existing studies, we noted that the sea level south of Tōkai was affected by the presence of large meanders in a broader sense, including atypical meanderings that straddle the Izu-Ogasawara Ridge. In the Atlantic, we found that the variability of the second mode drives the coherence across Cape Hatteras. We showed that

the strong link of this mode with the shifts of the Gulf Stream Extension west of 69°W is relatively recent and does not extend prior to ∼1990. We also showed that the local alongshore wind was an unlikely driver of this mode variability. Hence, whether this Atlantic second mode represents the pursuance of the same physical phenomenon, or if a mechanism supplanted another around ∼1990 is still an open question.

Because the tide gauge networks in both oceans extend further back in time than the period analysed in this study, inshore

sea level has potential for reconstruction of the variability of the ocean circulation mode of variability. Although the causal link between the upstream sea level and the meridional shifts of WBC extensions is not yet completely understood, our results suggest that upstream inshore tide gauges, such as Key West (available from 1913 in the PSMSL revised local reference (RLR) database), Fernandina Beach (1897) or Hosojima (1930) could be used as proxies for the extension meridional shifts and, by extension, the forcing responsible for such meridional shifts. In the Pacific, tide gauges in the region west of Kii, where the sea

level is less affected during large meander periods, should be preferred.

Reconstruction of past large meander events requires at least two gauges either side of the Cape Shiono-Misaki on the Kii peninsula and traditionally, the tide gauges of Uragami and Kushimoto are used. The two stations are available respectively from 1965 and 1957 in the PSMSL RLR catalogue. Using sea-level aggregates based on the correlation groupings (Figure 3) rather than the two individual tide gauge records, in a manner akin to McCarthy et al. (2015), allows the extension of the

analysis up to 1944 and characterize 1953 – 1955 and 1959 – 1963 as large meander periods, similar to Moriyasu (1958, 1961) and Kawabe (1985, 1995, 2005). Finally, further understanding of the forcing on sea level prior to ∼1990 is needed to use the north of Cape Hatteras gauge records as proxy for ocean and/or atmosphere variability prior to that date.

*Data availability.* This study GSNW and KE indices are available online, as are the principal components $\alpha_1$'s and $\alpha_2$'s. They can be downloaded in '.csv' spreadsheet format at https://doi.org/10.5281/zenodo.4659318 (Diabaté et al., 2021). When using these timeseries,





please cite the present study appropriately. The GSNW and KE indices are derived from the EN4 quality controlled ocean temperature profiles (Good et al., 2013, EN4.2.1) available at https://www.metoffice.gov.uk/hadobs/en4/, and from the $1981 - 2010$ objectively analysed mean temperature of the World Ocean Atlas 2018 (Locarnini et al., 2018, WOA18) available at https://www.nodc.noaa.gov/OC5/woa18/. Underlying datasets for $\alpha_1$ and $\alpha_2$ include (1) the original monthly tide gauge data available from the Permanent Service for Mean Sea Level (PSMSL, https://www.psmsl.org/), (2) sea-level pressure and ten meters above sea level wind speeds from the NCEP/NCAR Reanalysis 1

(Kalnay et al., 1996), distributed by the NOAA/OAR/ESRL PSL, and (3) gridded monthly SSH from the ARMOR3D product (Guinehut et al., 2012) available from the Copernicus Marine Environment Monitoring Service (CMEMS, https://marine.copernicus.eu).

## Appendix A: Adjustment of tide gauge records for surge-driven variability

Following the work of Dangendorf et al. (2013, 2014), Frederikse et al. (2017) and Piecuch et al. (2019), we compute the inverse barometer and wind-surge contributions on sea level using multiple linear regression, with pressure ($p$), alongshore

($\tau_\parallel$) and across-shore ($\tau_\perp$) wind stress anomalies interpolated at each tide gauge location as predictors. The angles used for the rotation in across-shore/alongshore coordinates are presented in Supplementary Table S1 and Supplementary Table S2. The quality of a regression is primarily depending of the correlation between the explanatory variables and of the period width. Here, the pressure and winds are not independent. Hence, for each tide gauge, an *all possible* regression procedure was designed. This means that, the model with the three regressors is tested (equation A1), alongside all the possible models where

first, second and (or) third term of that equations are (is) ruled out. In total, $2^3 - 1 = 7$ combinations of possible regression are tested at each tide gauge.

$$\zeta = -\beta_1 \left[ p(t) - \overline{p}(t) \right] + \beta_2 \tau_\parallel(t) + \beta_3 \tau_\perp(t) + \mathcal{O}(t). \tag{A1}$$

The regressions returns the regression coefficients $\beta_1$, $\beta_2$, and $\beta_3$, that describe the relationship between pressure, alongshore and across-shore winds and the gauge record. $\mathcal{O}(t)$ represents the unexplained residual. Y-intercepts are estimated to improve

the computation, but are not removed to form the residual $\mathcal{O}(t)$. Note that the inverse barometer effect is not proportional to the local pressure alone but to the difference between the local pressure and the global sea-level pressure $\overline{p}(t)$ averaged over the oceans. Also, $\beta_1$ is preceded by a minus ($-$) because rise of local atmospheric pressure makes sea level fall and vice versa.

     To determine the best model for each tide gauge, $95\%$ confidence intervals are computed for each regression coefficient within the Matlab built-in function. The regression models that feature one (or several) coefficient confidence interval(s) cross-

ing zero are excluded. Then, the best regression is defined as the one with the highest adjusted coefficient of determination $R^2_{Adj}$, which is the proportion of the variance in the gauge record that is predictable from the atmospheric predictors, adjusted for their number (see equation A2).

$$R^2_{Adj} = 1 - \frac{\sum\limits_{i=1}^{m} \left( \zeta_i - \widehat{Y}_i \right)^2}{\sum\limits_{i=1}^{m} \left( \zeta_i - \overline{\zeta} \right)^2} \frac{m-1}{m - n_r - 1}, \tag{A2}$$

where $m$ is the number of timestep, $n_r$ is the number of regressors for that particular model ($n_r \in [1, 2, 3]$), and $\widehat{Y}$ the sum of

the obtained atmospheric contributions.



Supplementary Figures S6 (a) and (b) present, respectively for the Atlantic and for the Pacific gauges, the regression coefficients $\beta_1$, $\beta_2$, and $\beta_3$ as well as their 95% confidence interval. For each gauge, only the output of the best regressions is shown. Hence, the number of regressors is not always three, in particular for the Japanese stations. The adjusted coefficients of determination $R^2_{Adj}$ (bottom) for each tide gauge are shown on Supplementary Figures S7.

We compute values for $\beta_1$ of 0.11 mm/Pa and 0.9 mm/Pa on average for American and Japanese gauges respectively (see green line on Supplementary Figures S6 (a) and (b) . In both regions, the computed $\beta_1$ are comparable to the theoretical $(\rho g)^{-1} \sim 0.10$ mm/Pa expected for an inverse barometer response (dashed green line). The pressure is always one of the explanatory variables of the best regression, highlighting the importance of the inverse barometer effect on sea level. Across-shore coefficients are high in locations upstream of the estuaries of Delaware Bay and Chesapeake Bay (TG 7, 8, 9, 10 and

12: Washington D.C., Solomon's Island, Annapolis, Baltimore and Philadelphia). There, the wind set up is amplified by the funnelling effect of the estuaries. The nearby stations of Sewells Point (TG 6) and Lewes (12) are, in contrast, much less affected, because they are located at the mouth of the estuaries. The best regression does not feature the across-shore wind north of Cape Cod (TG 20 – 22). The obtained alongshore coefficients show less deviations from the average of $-7.9$ m$^3$/N, yet the maximums are also found in the estuaries region. Finally, it is to note that, north of Cape Cod (TG 14 to 23), we

obtain values for $\beta_2$ an order of magnitude greater than reported by Piecuch et al. (2019). For most of the Japanese gauges, the best regression does not feature either the across-shore or the alongshore component of the wind stress, as using all of the explanatory variables does not explain more variability in the tide gauge records. However a consistent effect of the alongshore winds for the southern tide gauges (TG 1 – 2 and 4 – 19) is revealed by the regression.

Supplementary Figure S7 present the adjusted coefficient of determination $R^2_{Adj}$ (Eq. A2). It depicts better the effect of the

atmosphere on the sea level than the regression coefficients alone, as even a weak $\beta_i$ coefficient could affect greatly the sea level if the corresponding regressor variability is important at the tide gauge location. Consistently with Piecuch and Ponte (2015), we find that the atmospheric effect on sea level explains an important part of the gauge variability north of Cape Hatteras (Supplementary Figure S7 (a)), with $R^2_{Adj}$ on average superior to 40%. This result is in agreement with Piecuch et al. (2019), who reported a value of 39%, although their analysis focussed on the period 2004 – 2017. In contrast, only 20% of

the variability in the timeseries of the southern gauges can be explained with the regression, with some deviations from that mean. This is consistent with previous findings of Woodworth et al. (2014) and we observe a similar pattern in the Pacific. The tide gauges located north of the Bōsō Peninsula (Supplementary Figure S7 (b)) show high $R^2_{Adj}$, whereas tide gauges south of Japan (TG 1 to 24) are not at all explained by the atmospheric forcing. The drop in the variability explained by the atmosphere is, in both regions, located at the separation point of the western boundary current (Cape Hatteras and the Bōsō Peninsula

respectively). This does not necessarily means that there is no atmosphere-related sea level change south of the separation points but rather that they are dwarfed by other sources of variability.

Supplementary Figures S8 (a) and (b) present the mean of $\zeta_{Atm}$ north of the Bōsō Peninsula and Cape Hatteras respectively, where $\zeta_{Atm}$ is the sea-level driven by the atmosphere and regroups the three first terms of equation A1 ($\zeta = \zeta_{Atm} + \mathcal{O}(t)$). It is apparent that, while most of the variability is intra-annual, there is also interannual variations.





In the paper, we consider the residual $\mathcal{O}(t)$ (Eq. A1), which represents the sea-level variability unexplained by the atmospheric variables, as the sea-level 'corrected' from the local atmospheric forcing, and refer to it as $\zeta$ for simplicity.

**Appendix B:  Empirical Orthogonal Function Analysis**

Here we provide further insights on the Empirical Orthogonal Function (EOF) analysis used to objectively reduce the sea-level anomalies in an ensemble of modes.

EOF decomposition using covariance matrix is performed on tide gauge records after they are interpolated on a regular grid to prevent variability in better sampled region from being favored in the analysis. First, tide gauges located in the Chesapeake Bay, the Inland Sea (west and east separately) are averaged and associated with a virtual location at the mouth of, respectively, the Chesapeake Bay, the Bungo Channel and the Kii Channel. Then, in the Pacific (Atlantic), the 20 (18) remaining gauge records plus the two (one) aggregated estuary records are interpolated onto a regular, alongshore grid with 150 km spacing.

The modes are composed of a time-varying coefficient $\alpha$, the *Principal Component* (PC), and an associated spatial-varying coefficients $\phi$, the *Empirical Orthogonal Vector* or *Function* (EOF):

$$\zeta(x,t) = \sum_{i=1}^{n} \alpha_i(t)\phi_i(x),\qquad\qquad(\text{B1})$$

where $i = 1, 2, 3, ...$ represents the modes, which are ordered by decreasing percentages of total variability explained, $n$ is the total number of spatial grid points and $x$ and $t$ are, respectively, space (alongshore) and time.

Note that the spatial amplitudes $\phi$'s that are discussed within the text are relative to periods when their associated principal component $\alpha$'s are equal to standard deviation, that is, $\alpha_i = 1$. For example, it is shown that the amplitude of the leading mode south of Japan, $\phi_1$, is on average 2.3 cm (Fig. 5 (a)). The peak-to-peak amplitude of the associated temporal amplitude $\alpha_1$, defined as the difference between the maxima of late 2004 and the minimum of 1985, is roughly equal to 5 (Fig. 5 (c)). Hence, between 1985 and 2004, the sea-level rise associated with this mode is of $\sim 12$ cm south of Japan.

*Author contributions.* Samuel Diabaté set up the methodology, and carried out the investigation and the analysis. Gerard McCarthy and Didier Swingedouw supervised the research and conceptualized research goals. Samuel Diabaté, Ivan Haigh and Gerard McCarthy implemented the computer code for analysis and visualization. Samuel Diabaté and Gerard McCarthy prepared the manuscript with contributions from all co-authors.

*Competing interests.* The authors declare that they have no conflict of interest.

*Acknowledgements.* Samuel Diabaté would like to thank his colleagues of the A4 team as well as Benoit Meyssignac, Simon Michel, Juliette Mignot and David Pugh their valuable advice and suggestions ; Norihisa Usui, Magdalena Andres, Christopher Piecuch, Arnoldo



Valle-Levinson, Dudley Chelton, Thomas Frederikse, Karen Simon, Philip Woodworth and David Smeed for their mail correspondence ; and Jeanne Auboiron for her preliminary work.

Gerard McCarthy and Samuel Diabaté work as part of the A4 Project (Aigéin, Aeráid, agus Athrú Atlantaigh — Oceans, Climate, and
Atlantic Change ; Grant-Aid Agreement No. PBA/CC/18/01), which is carried out with the support of the Irish Marine Institute under the Marine Research Programme funded by the Irish Government, co-financed by the European Regional Development Fund. Gerard McCarthy is further supported by the ROADMAP project (Grant-Aid Agreement No. PBA/CC/20/01) supported by the Marine Institute and funded by the Irish Government under the 2019 JPI Climate and JPI Oceans Joint Call. Didier Swingedouw received support from the Blue-Action (European Union's Horizon 2020 research and innovation program, Grant Number: 727852) and EUCP (European Union's Horizon 2020
research and innovation programme under Grant Agreement No. 776613) projects. Joël Hirschi acknowledges funding from the Newton Fund CSSP China project DYVA and from the NERC project ACSIS (NE/N018044/1).





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
