# Peer review of "Western boundary circulation and coastal sea-level variability in northern hemisphere oceans"

_Ocean Science, 2021_

## Referee Comment (RC3)

**Comments on "Western boundary circulation and coastal sea-level variability in northern hemisphere oceans" by Diabate et al, submitted to Ocean Science Discussions (https://doi.org/10.5194/os-2021-24)**

Tal Ezer
Center for Coastal Physical Oceanography, Old Dominion University, Norfolk VA, USA

**General comments:**

The paper compares coastal sea level variability near two WBCs, the Kuroshio and the Gulf Stream (GS), and finds interesting links between sea level and shifts in the position of the currents after they separated from the coast. EOF analysis highlights some similarities and some differences between the two regions and emphasizes the different sea level response north and south of the separation point. The topic of remote influences on coastal sea level and links to open ocean currents is an important new area of research in recent years, and this study is certainly a nice additional contribution, even though some of the results may not be completely new. The paper is well written and the results worth publication, though it could be improved by having more clear indication what is new or novel here relative to other recent studies of WBCs (some not cited, see below).

**Major comments:**

1. One notable result, the dramatic shift in the North Atlantic region after the 1990s (Figs. 4 and 7) is not completely explained, in my opinion. Several recent papers on the topic that were not cited can shed more light on the topic and help the authors in their explanation of the dynamics involved. For example, Chen et al. (2019) and Ezer and Dangendorf (2021) already compared WBCs dynamics including the Kuroshio and the Gulf Stream, both discuss the link between uneven warming, spatial sea level rise differences and the intensity of WBCs (little is said in the current paper about the role of temperature change). While they did not focus on meridional shifts of WBCs, the position and intensity of WBCs are closely related, and this point is somewhat lost. On decadal time scales recent studies show that the Kuroshio is correlated with the wind much more significantly than the GS does (see Fig. 11 in Ezer & Dangendorf, 2021), which may partly explain why abrupt changes in the GS due to internal dynamics may be more likely. On decadal time scales there is also a shift in the GS transport from strengthening during 1970-1990 to weakening in 1990-2015 (see Fig. 9 in Ezer and Dangendorf, 2020) - was this shift in transport related to the change you found around 1990?

Another topic that was not fully explored is the disconnect north and south of the separation point (Fig. 3), which was investigated by others - a recent paper (Ezer, 2019, not cited) fucus on this very point, suggesting an explanation related to the proximity of the GS to the coast, showing how propagation of positive temperature anomalies can cause coastal sea level rise/fall for locations south/north of the separation point, as the GS intensified and shifts after separation, but stays near the coast with warmer waters in the south. I hope all these new studies will help the authors putting their results in context with other recent findings.

Chen C, Wang G, Xie S, Liu W (2019) Why does global warming weaken the Gulf Stream but intensify the Kuroshio? J. Clim. 32:7437–7451. https://doi.org/10.1175/JCLI-D-18-0895.1

Ezer, T., (2019), Regional differences in sea level rise between the Mid-Atlantic Bight and the South Atlantic Bight: Is the Gulf Stream to blame?, Earth's Future, 7(7), 771-783, doi:10.1029/2019EF001174.

Ezer, T. and S. Dangendorf (2020), Global sea level reconstruction for 1900-2015 reveals regional variability in ocean dynamics and an unprecedented long weakening in the Gulf Stream flow since the 1990s, Ocean Science, 16(4), 997-1016, doi:10.5194/os-16-997-2020.

Ezer, T. and S. Dangendorf (2021), Variability and upward trend in the kinetic energy of Western Boundary Currents over the last century: impacts from barystatic and dynamic sea level change, Climate Dynamics, doi:10.1007/s00382-021-05808-7.

**Specific comments and suggestions:**

2. Abstract, line 6: "… comparison between the two basins is missing", the statement is not completely true given the recent studies mentioned above that compared the two WBCs (e.g., Chen et al., 2019; Ezer and Dangendorf, 2021; others). This fact could be added to the introduction.

3. Lines 59-64: in addition to the cited Andres et al. (2013), please take a look at a more recent paper from the same group (Andres et al., 2020), which studied the path of the GS at two sections. They showed large differences between the western GS affected by local recirculation and the eastern GS where variations in the meridional path are much larger. The large spatial differences over short distances along the GS may explain some of the discrepancies you cited.

Andres, M., Donohue, K. A., and Toole, J. M. (2020), The Gulf Stream's path and time-averaged velocity structure and transport at 68.5W and 70.3W, Deep-Sea Res. Pt. I, 156, 103179, https://doi.org/10.1016/j.dsr.2019.103179, 2020.

4. Lines 85-89: The increase in kinetic energy of WBCs (Ezer and Dangendorf, 2021) and differences between the Kuroshio and GS with respect to AMOC (Chen et al., 2019) may be relevant to add here.

5. Lines 276-277: in addition to McCarthy et al. (2015) and Woodworth et al. (2014), the more recent study of Ezer (2019) focus specifically on the drop in correlation north and south of Cape Hatteras (in fact, Fig. 2 in Ezer's paper using altimeter data is the equivalent to and confirmation of Fig. 3 here using tide gauge data).

6. Lines 292-295: is it possible that the drop in correlation in the Atlantic may relate to weakening GS at that period (Ezer and Dangendorf, 2020) or/and to the shift in the hotspot of sea level rise (Valle-Levinson et al., 2017; Ezer, 2019)? Dynamically, it makes sense that a stronger current in early years has more coherence along its path than a weaker one that can be affected more easily by local factors. Have you considered this option?

7. Line 355: there is no "Figure 9", should be 6?

8. Line 469: "… re-qualified the Gulf Stream presence as a plausible sea-level driver…", it may be clarified that this result is not new and confirms many previous studies (e.g., Ezer et al., 2013, Ezer, 2013, 2015, 2019).

9. Lines 532-541: the "puzzling" result of the shift around 1990 is still not clearly explained, with several hypotheses offered. It may also be useful to see if this shift links to recent weakening of the GS and the southward shift in the hotspot of SLR from north of Cape Hatteras to south of Cape Hatteras; see comments above and Valle-Levinson et al. (2017) and Ezer (2019).

10. Line 641: "… further understanding of the forcing on sea level prior to ~1990 is needed…", it may be useful to add here that studying the link between open ocean dynamics and variations in coastal sea level in early years is possible and has been recently done using reconstructed sea level

approaches going back to 1900; see for example studies such as Ezer and Dangendorf (2020) and Dangendorf et al. (2021).

Dangendorf, S., T. Frederikse, L. Chafik, J. Klinck, T. Ezer, and B. Hamlington, (2021), Data-driven reconstruction reveals large-scale ocean circulation control on coastal sea level, Nature Climate Change, doi:10.1038/s41558-021-01046-1

---

## Community Comment (CC1)

I want to thank Anonymous Referee #1 for their comments. They are greatly appreciated and will help improve the manuscript.

**Comment 1:**

"The manuscript is well written, but the novelty of this study is unclear. In the abstract, there are only two sentences about the results of the present study (L6-10). As the authors cited in the manuscript, there are many studies that examined coastal sea level variability associated with the Kuroshio and the Gulf Stream variability. Also, recent review paper (Woodworth et al. 2019) compared coastal sea level variability between the Kuroshio and the Gulf Stream regions. What are the new findings in the present study? Please more clarify this point."

**Response:**

In the Pacific, the coastal sea level upstream of the Kuroshio has been shown to co-variate with, on the one hand, the location of the Kuroshio as it approaches the lzu-Ogasawara Ridge (Kuroda et al., 2010), and on the other hand, with the atmospheric regime shifts in central North Pacific (Senjyu et al., 1999). These different results have been nicely tied up together by Sasaki et al. (2014) and Yasuda and Sakurai (2006), who have argued that Rossby wave, breaking as coastally trapped wave at arrival at the western margin, bring the central Pacific signal to the coast of Japan, while modifying the latitude of the Kuroshio. Our results compliment these findings, as we show using altimetry and subsurface temperature that the main mode of sea-level variability reflects change in the Kuroshio Extension location. However, findings of Sasaki et al. (2014) were limited to 1993 – 2011, and those of Yasuda and Sakurai (2006) to the model world, whereas our results hold for 1968 – 2019 and are solely based on observations.

In the Atlantic, to the contrary—and to the best of my knowledge—no study (including the recent work of Woodworth et al. 2019) had previously associated the location of the Gulf Stream Extension with the upstream sea level. That the agreement between the upstream sea level and the meridional shifts of the extension holds in both ocean separately is a new result, and our main finding. This finding is new and of great importance for the community interested in the sea-level variability of the North Atlantic western margin, and to those interested in the Gulf Stream North Wall, while in the same time it brings further evidence to the community working on the Japanese costal sea level and/or on the Kuroshio Extension.

We understand from the comment of Anonymous Referee #1 that the novelty of the study is not straightforwardly shown in the manuscript. We have identified places where the manuscript could be modified (abstract, introduction, conclusion) to present more accurately than at present what has been summarised in the two above paragraphs. An improved version of the manuscript will be presented in the final response.

**Comment 2:**

"As the authors pointed out in section 3.2.1, the sea surface velocity changes associated with the first EOF mode in the upstream region of the separation point is different between the North Atlantic and North Atlantic (L341-349). The Kuroshio was shifted on-shoreward, but the Gulf Stream was shifted off-shoreward in the positive phase of the first EOF mode (Fig. 5), although the corresponding costal sea level anomalies are positive. However, the detailed difference has not been discussed in the manuscript. It is interesting to compare the sea level change in the across-shore direction between the two regions."

**Response:**

Indeed, the upstream patterns of sea surface velocity are different in each basin. If significant, this discrepancy could be due to different interactions in each basin between the upstream jet and the bathymetry. It suggests that some more involved mechanisms than a pure Kelvin wave are at work. Across-jet analysis of the sea surface height (SSH) variability is feasible for the region southeast of Japan, because, there, the lateral shifts of the jet are large (Figure 5.a of the manuscript). Figure 1 below shows such analysis, where the 138.875°E was used. Agreement between the leading principal component of the tide-gauge obtained sea-level anomaly and the meridional shifts of the Kuroshio at 138.875°E is visible. Overall this analysis produces the same results than the sea surface velocity (SSV) analysis presented in the manuscript, and no different conclusion can be made.

For the upstream Atlantic situation, an across-jet SSH analysis does not produce satisfactory results and is hence not shown. Comparatively to southeast Japan, where the change of path of the Kuroshio produces large SSV change (Figure 5.a of the manuscript), SSV changes across the Florida Current are small and have small zonal extent (see Figure 5.b of the manuscript), which largely complicates an across-jet analysis. Nonetheless, that the Florida Current shifts off-shoreward in the positive phase of the first EOF mode is not impossible. A similar situation is seen in the Pacific, southeast of Kyūshū (Figure 5.a of the manuscript). Indeed, when the Kuroshio shifts off-shoreward at 138.875°E, it moves in-shoreward southeast of Kyūshū (See Figure 2 below). The upstream situation in the Atlantic hence doesn't necessarily conflicts with the patterns seen in the Pacific. In any case, and most importantly, conclusions drawn from SSH analysis, as those obtained with SSV analysis, are only valid from 1993 onwards due to the altimetry data availability. Contrarily to the region of the Gulf Stream and Kuroshio extensions, we cannot validate altimetry obtained results for the upstream Gulf Stream and Kuroshio with subsurface temperature indices extending further back in time. Hence, this topic was not developed furthermore in the manuscript.

We understand from the comment of Anonymous Referee #1 that a more developed discussion on the behaviour of the upstream jet would be beneficial to the paper. While I agree with Anonymous Referee #1, I must also admit that what we can bring to the discussion is limited, because the altimetry results for the upstream situation cannot be cross-validated with subsurface temperature (at least, not with the data we have produced). A detailed investigation of the difference between the two upstream situation (Pacific and Atlantic) and of the involved mechanisms is beyond the scope of this work. Nonetheless, any changes to the manuscript will be presented in the final response. Any advices on this very interesting subject are welcome.

Again, I want to thank Anonymous Referee #1 for their comments.

Samuel Tiéfolo Diabaté samuel.diabate.2020@mumail.ie

**Figure 1:** Change in SSH in the across-jet direction at 138.875°E (South of Japan, just west of the Izu-Ogasawara Ridge). (a) Hovmöller diagram of change in SSH along the 138.875°E. The thin black line represents the 1.05-meter iso-SSH. (b) Variations in time of the 1.05-meter iso-SSH at 138.875°E (blue), and same quantity after 19-month filtering (red). (c) The filtered 1.05-meter iso-SSH at 138.875°E (red) next to the leading principal component of tide-gauge obtained sea-level anomaly (blue). The 138.875°E longitude is close enough to the ridge for the SSH variability to be only weakly affected by occurrence of the typical Large Meander—atypical meandering (straddling the ridge) is however apparent in 2000 – 2001, and explains the disagreement with the leading principal component of the sea-level anomaly then (Panel c).

---

## Author Comment (AC1)

Dear anonymous referee #1,

Following our previous answer, we present in this response the changes we made to the manuscript. The detailed changes are appended at the end of this document.

**Novelty**
Establishing the novelty of this study has now been done in the manuscript. The abstract was slightly changed to state that this study extend previous results limited to the altimetry era. The introduction, discussion and conclusion were changed to contextualize further the study with the references you suggested, particularly with Ezer (2019) and Dangendorf et al. (2021) since these are the most relevant. Also, greater emphasize was given to our main result, which is that there is an agreement between the upstream sea level and the meridional shifts of the WBC extensions in both ocean separately, a result that was previously only shown for the altimetry era.

**Upstream patterns associated with the leading principal components**
The SSV patterns associated with the leading principal components were different in the two basins. The Kuroshio was shifted on-shoreward, but the Gulf Stream was shifted off-shoreward in the positive phase of the first mode. You suggested we investigate the subject more.

A paragraph was added to the discussion which states that off-shoreward motion during positive phase of the mode is also seen in the Pacific (southeast of Kyushu) so that the situations in the two basins are not necessarily contradictory (see our previous author's response). We also develop on the role of oNLM and nNLM path alternance and of temperature changes. A paragraph that used to be in the discussion and briefly mentioned the upstream patterns was reformulated and moved to the conclusion.

Finally, investigating this subject has led me to question whether the upstream patterns seen in the Atlantic were due to the Ekman component of the sea surface velocities of the Rio et al (2014) product. The short answer is that it is not. None of the patterns seen in Figure 5 and 6 of the manuscript are greatly changed when using geostrophic dataset which do not contain an Ekman component (like the ARMOR3D dataset, Mulet et al., 2012). In Figure 1 of the present document (below) is shown a comparison between the composites obtained for Figure 5 of the manuscript. In order to remove any possible confusion for the reader, the figures in the manuscripts are now all produced with the ARMOR3D dataset, which is purely geostrophic, and the manuscript was updated accordingly.

Again, I want to thank you for your comments and the time invested.
Best Regards,

Samuel Tiéfolo Diabaté, on behalf of all co-authors
samuel.diabate.2020@mumail.ie

[Figure]

Figure 1 – The composite obtained with the ARMOR3D surface velocity product (no Ekman component) (left column) and Rio et al. (2012) surface velocity product (contains an Ekman component) (right column). The difference between the two are extremely small.

[revised manuscript text omitted]

---

## Author Comment (AC2)

Dear Tal Ezer,

I want to thank you on behalf of all authors for your comments. They are greatly appreciated and will help improve the manuscript.

**1) Novelty of the study**

**General comment**

> "The paper compares coastal sea level variability near two WBCs, the Kuroshio and the Gulf Stream (GS), and finds interesting links between sea level and shifts in the position of the currents after they separated from the coast. EOF analysis highlights some similarities and some differences between the two regions and emphasizes the different sea level response north and south of the separation point. The topic of remote influences on coastal sea level and links to open ocean currents is an important new area of research in recent years, and this study is certainly a nice additional contribution, even though some of the results may not be completely new. The paper is well written and the results worth publication, though it could be improved by having more clear indication what is new or novel here relative to other recent studies of WBCs (some not cited, see below)."

**Response**

As anonymous referee #1, you have indicated that the novelty of our study is not very clear in the manuscript. Our main finding is that there is an agreement between the upstream sea level and the meridional shifts of the WBC extensions in both ocean separately. This agreement holds for the totality of the period studied, that is, the past seven (five) decades in the Atlantic (Pacific). This is a new result for the Pacific basin in the sense that the similar findings of Sasaki et al. (2014) were limited to 1993 – 2011. In the Atlantic, no study had previously demonstrated the agreement between the variability of the upstream sea level (south of Cape Hatteras) and the evolution of the Gulf Stream Extension position[1], at two notable and rather recent exception. Dangendorf et al. (2021) (that you mentioned) showed that the stero-dynamic portion of the upstream tide gauge variability was correlated with the steric height signal in the Gulf Stream Extension, and hinted for a role of *'subtle variations in the strength and position of the Gulf Stream'*. In practice this excellent paper was made available online after our initial submission, reason why there was no mention of it in our manuscript. In Ezer (2019) (that you mentioned, also), it is shown that the upstream sea level and the Gulf Stream Extension position covary at decadal to multidecadal timescale. Again, this finding was limited to the altimetry era, whereas our result is true for a much longer period.

**Changes in the manuscript**

Establishing the novelty of this study has now been done in the manuscript. The abstract was slightly changed to state that this study extend previous results limited to the altimetry era. The introduction, discussion and conclusion were changed to contextualize further the study with the references you suggested, particularly with Ezer (2019) and Dangendorf et al. (2021) since these are the most relevant. Also, greater emphasize was given to our main result.

The change to the manuscripts are shown in the pages appended at the end of the present document.
* * *
[1] In fact, the Gulf Stream Extension position was more often associated with the sea-level north of Cape Hatteras in the Mid-Atlantic Bight (Ezer et al., 2013; Andres et al., 2013), and we describe in the manuscript how this agreement is recent (since circa 1990).

**2) Role of volume transport (and temperature changes), decorrelation around 1990, and variance change in the Atlantic second mode.**

In this section author, we address in particular one of your major remark, which is, that we discussed little the roles of volume transport and of temperature in the variability of the sea-level modes we obtained, nor in the sea-level variability decorrelation north and south of the separation point in the Atlantic around 1990.

**Major comments:**

"1. One notable result, the dramatic shift in the North Atlantic region after the 1990s (Figs. 4 and 7) is not completely explained, in my opinion. Several recent papers on the topic that were not cited can shed more light on the topic and help the authors in their explanation of the dynamics involved. For example, Chen et al. (2019) and Ezer and Dangendorf (2021) already compared WBCs dynamics including the Kuroshio and the Gulf Stream, both discuss the link between uneven warming, spatial sea level rise differences and the intensity of WBCs (little is said in the current paper about the role of temperature change). While they did not focus on meridional shifts of WBCs, the position and intensity of WBCs are closely related, and this point is somewhat lost. On decadal time scales recent studies show that the Kuroshio is correlated with the wind much more significantly than the GS does (see Fig. 11 in Ezer & Dangendorf, 2021), which may partly explain why abrupt changes in the GS due to internal dynamics may be more likely. On decadal time scales there is also a shift in the GS transport from strengthening during 1970-1990 to weakening in 1990-2015 (see Fig. 9 in Ezer and Dangendorf, 2020) - was this shift in transport related to the change you found around 1990?
Another topic that was not fully explored is the disconnect north and south of the separation point (Fig. 3), which was investigated by others - a recent paper (Ezer, 2019, not cited) fucus on this very point, suggesting an explanation related to the proximity of the GS to the coast, showing how propagation of positive temperature anomalies can cause coastal sea level rise/fall for locations south/north of the separation point, as the GS intensified and shifts after separation, but stays near the coast with warmer waters in the south. I hope all these new studies will help the authors putting their results in context with other recent findings."

The specific comments 6 and 9 are also linked to this topic, to they are included here:

"6. Lines 292-295: is it possible that the drop in correlation in the Atlantic may relate to weakening GS at that period (Ezer and Dangendorf, 2020) or/and to the shift in the hotspot of sea level rise (Valle- Levinson et al., 2017; Ezer, 2019)? Dynamically, it makes sense that a stronger current in early years has more coherence along its path than a weaker one that can be affected more easily by local factors. Have you considered this option?"

"9. Lines 532-541: the "puzzling" result of the shift around 1990 is still not clearly explained, with several hypotheses offered. It may also be useful to see if this shift links to recent weakening of the GS and the southward shift in the hotspot of SLR from north of Cape Hatteras to south of Cape Hatteras; see comments above and Valle-Levinson et al. (2017) and Ezer (2019)."

**Response**

We investigated the role of volume transport of the Kuroshio and of the Gulf Stream, but eventually decided to not include this part in the submitted manuscript, because the results were

not very conclusive and in an effort to shorten the paper. In particular, we investigated the agreement between the leading principal components of the sea-level modes and the WBC transport using the volume transport measured by submarine cable between the Bahamas and Florida at 27N and Wei et al. (2013) estimate of the Kuroshio volume transport based on regular hydrographic sections at PN line (yellow lines on Figure 1 below). We find absence of correlation in the Atlantic (r = -0.04). The situation in the Pacific is more complex, with clear non-stationary in the coherence between the transport and the leading principal component of the sea-level mode. Indeed, the 1998-2006 features strong anti-variations, but because there is very little agreement over the remaining periods, there is only weak anticorrelation on average (r = -0.26). Hence, it is difficult to associate the sea-level variability of those leading modes to the volume transport directly.

You also pointed out that the change in the coherence between south and north of Cape Hatteras occurring around 1990 could be attributed to volume transport, or that, at least, this possibility should have been further discussed. This is undeniably an interesting question. The 2D matrix of the alongshore sea-level variability can be reconstructed as the sum of the obtained EOFs times their associated PCs. Taken together, the two leading modes explain 85% of the variance of the dataset and are well able to replicate the sea-level variability all along the coast (see Figure 2 below), including the change in coherence across Cape Hatteras. In fact, EOF analysis reconstructs the change in correlation across Cape Hatteras through an increase in the second mode variance (Figure 7a of the manuscript). Indeed, the second mode impacts more strongly the northern tide gauges than the southern, which implies decorrelation in sea-level variability across Cape Hatteras when the mode's principal component variance increases (after 1990).

The EOF analysis hence leads to interpret that the change in coherence across Cape Hatteras is due to a continuous mechanism (represented by the second EOF mode) whose effects were felt more strongly after 1990. Or at least, it does if we fully trust the EOF analysis to produce *physical* modes. However the agreement of the second principal component with both the Gulf Stream position and with atmospheric variables is seen only post ~1990, after the increase in the mode variance (Figure 7b of the manuscript and Figure S5 of the supplementary material). Hence, we weren't able to link the second mode with a continuous physical mechanism. It is thus possible that this EOF-PC couple is a *statistical* rather than a *physical* mode. If a physical mechanism replaced another around 1990, for example and as you suggested, under the background influence of Gulf Stream transport changes or AMOC changes, it is likely that EOF analysis would not distinguish between the two mechanisms and mix them into a single statistical mode. That is because EOF analysis is not designed to deal with physical mechanisms whose spatial footprints are non-stationary in time (such as disappearing or emerging modes).

I therefore agree with you that the disconnect north and south of the separation point is not fully explained. It is not clear if a physical mechanism with increasing effects caused the change in correlation pattern, or if a new mechanism appeared circa 1990, which cannot be resolved by EOF analysis. The volume transport estimate you referred to (Ezer and Dangendorf, 2020) doesn't seem to show a changing point around 1990, so I don't think it is relevant to include it here.

In Ezer (2019), you also hypothesized that a warming signal in the in the Sargasso Sea south of 35N and west of 60W may enhance the Gulf Stream (zonal) transport after the separation point (in the MAB region) through thermal wind balance, but may cross the Florida current and rise the sea level of the South Atlantic Bight coasts. Because Gulf Stream transport in the MAB and Gulf Stream Extension position are connected at low frequency (Ezer, 2019), this would be a link between upstream sea-level and WBC extension and an alternative to the Sasaki et al. (2014) proposition. However, in Dangendorf et al (2021), that you co-authored, Extended Data Fig. 3.c

does not show any agreement between the steric component of coastal sea-level in the SAB and the steric component of sea-level in the Sargasso Sea. There is no clear sign of a thermosteric signal propagating across the Florida Current there (or perhaps this mechanism is non-stationary?).

**Changes in the manuscript**

A paragraph was added to the discussion. It summarizes what is said above on the limitations of stationary EOF analysis, and on the possibility of a role of AMOC or volume transport in the change around 1990. A sentence in the conclusion was also added.

There seems to be a role of the temperature in the variability of the upstream tide gauges (Domingues et al., 2018; Kuroda et al., 2010; Ezer, 2019), so the conclusion was re-worked following your suggestion. We are now less conclusive on the Sasaki framework than before. Reference was made first to Domingues et al., 2018 and to Kuroda et al., 2010, as they link the upstream sea-level variability with the WBC temperature. Then reference was made to other papers that link the upstream sea-level variability to the more broad western subtropical gyre temperature and/or SSH (including Ezer, 2019).

The changes made to the document can be found in the pages appended at the end of the present document.

**3) Specific comments and suggestions:**
Here we provide answers to the minor suggestions you made.

Comment:
> "2. Abstract, line 6: "… comparison between the two basins is missing", the statement is not completely true given the recent studies mentioned above that compared the two WBCs (e.g., Chen et al., 2019; Ezer and Dangendorf, 2021; others). This fact could be added to the introduction."

Response:
This is true. A short part was added to the introduction.

Comment:
> "3. Lines 59-64: in addition to the cited Andres et al. (2013), please take a look at a more recent paper from the same group (Andres et al., 2020), which studied the path of the GS at two sections. They showed large differences between the western GS affected by local recirculation and the eastern GS where variations in the meridional path are much larger. The large spatial differences over short distances along the GS may explain some of the discrepancies you cited."

Response:
I agree with this comment. A paragraph was added to state that different mechanisms forcing the position of the Gulf Stream might coexist at different downstream distances from Cape Hatteras, and partly explain the different behaviour along the jet.

Comment:

"4. Lines 85-89: The increase in kinetic energy of WBCs (Ezer and Dangendorf, 2021) and differences between the Kuroshio and GS with respect to AMOC (Chen et al., 2019) may be relevant to add here."

Response:
This was added to the text.

Comment:
"5. Lines 276-277: in addition to McCarthy et al. (2015) and Woodworth et al. (2014), the more recent study of Ezer (2019) focus specifically on the drop in correlation north and south of Cape Hatteras (in fact, Fig. 2 in Ezer's paper using altimeter data is the equivalent to and confirmation of Fig. 3 here using tide gauge data)."

Response:
This was added to the text.

Comment:
7. Line 355: there is no "Figure 9", should be 6?

Response:
That's correct. It should have been Figure 5. It was changed in the text.

Comment:
"8. Line 469: "… re-qualified the Gulf Stream presence as a plausible sea-level driver…", it may be clarified that this result is not new and confirms many previous studies (e.g., Ezer et al., 2013, Ezer, 2013, 2015, 2019)."

Response:
The sentence was changed to omit 're-qualified' which was an inappropriate term. We now simply state that our result suggests that the Gulf Stream presence is a plausible sea-level driver. The mention to the publications suggested was added.

Comment:
"10. Line 641: "… further understanding of the forcing on sea level prior to ~1990 is needed…", it may be useful to add here that studying the link between open ocean dynamics and variations in coastal sea level in early years is possible and has been recently done using reconstructed sea level approaches going back to 1900; see for example studies such as Ezer and Dangendorf (2020) and Dangendorf et al. (2021)."

Response:
A short sentence was added to the text.

The comments numbered 6 and 9 concerned the drop in correlation across Cape Hatteras and the change of variance in the Atlantic second mode. They were answered above in **2)**.

Finally, note that the SSV product used was changed and the discussion on the upstream SSV patterns of Figure 5 of the manuscript was enhance (cf. Last Author's Comment to Anonymous Referee #1).

Again, I want to thank you for your comments and the time invested.
Best Regards,

Samuel Tiéfolo Diabaté, on behalf of all co-authors
samuel.diabate.2020@mumail.ie

[Figure]

**Figure 1 -** The leading principal component and the WBC transport for the Pacific (a) and the Atlantic (b). Wei et al. (2013) estimate of the Kuroshio transport covers 1955--2010, but is constrained prior to 1987 by the climatological seasonal mean of the Tokara Strait transport. Hence, we only use the 1987 – 2010 estimate of the Kuroshio transport at PN line. Sometimes two cruises sampled the PN line within one season, and there is two estimates of the transport for a single timestep. In this case we averaged the two estimates. Wei et al. (2013) estimate is quaterly, so the correlation is computed after the principal component is quaterly averaged.

[Figure]

**Figure 2 – (a)** Mean of the sea level south of Cape Hatteras (thick violet line), together with reconstructed sea-level from the EOF modes 1 and 2 (orange and blue shades). **(b)** As for (a), but with the sea level north of Cape Hatteras. Mean of the sea level north of Cape Hatteras is in green.

[revised manuscript text omitted]

---

## Author Response (AR1)

**Author's response**

Both reviewers indicated that the novelty of our study is not very clear in the manuscript. Changes were made to the manuscript in order to improve it. The introduction, discussion and conclusion were changed to contextualize further the study with the references suggested by the reviewers, particularly with Ezer (2019) and Dangendorf et al. (2021) since these are the most relevant. The discussion and conclusion were particularly reworked, with greater emphasize now given to our main result. The abstract was slightly changed to state that this study extend previous results limited to the altimetry era.

Anonymous reviewer #1 suggested to discuss in more details the upstream patterns of sea surface velocity in both oceans, whereas Tal Ezer suggested to consider in the discussion the role of water temperature and transport. A paragraph was added to the discussion which states that off-shoreward motion during positive phase of the leading principal component is also seen in the Pacific (southeast of Kyushu) so that the situations in the two basins are not necessarily contradictory. We also develop on the role of oNLM and nNLM path alternance and of temperature changes in the discussion. A paragraph that used to be in the discussion and which briefly mentioned the upstream patterns was greatly enhanced and moved to the conclusion. Therefore the upstream situation is now much more discussed in the manuscript than before. The role of temperature is also further discussed. Reference was made to Domingues et al. (2018), Ezer (2019) and to Kuroda et al. (2010) as they link the upstream sea-level variability with the WBC temperature. Consideration of both the upstream situation and of the role of transport and temperature led us to be less conclusive on the Sasaki et al. (2014) framework than before.

All minor comments from the referees were accepted and subsequent changes were made to the manuscript.

Finally, additional changes were made when we thought they were needed. The sea surface velocity product used was changed, and the figures were printed again. This was made to avoid any confusion regarding the Ekman component of the dataset that was previously used. The dataset we now used has no Ekman component, which makes the manuscript easier to understand. Additionally, a small error in our script, which was leading to slightly incorrect isolines in Supplementary Figure S4, was corrected. The difference is hardly discernible to the naked eye. The Northern Recirculation Gyre line in Figure 1 was previously missing and was added. Ordinate and abscissa labels were added in Figures 3 and 4. The captions of Figures 5 and 6 were reworked to be more clear. A few typos were corrected. Finally, the last two paragraph were inverted to end the manuscript on a more positive note.

Samuel Diabaté, On behalf of all co-authors.

**References**

Dangendorf, S., Frederikse, T., Chafik, L., Klinck, J. M., Ezer, T., and Hamlington, B. D.: Data-driven reconstruction reveals large-scale840ocean circulation control on coastal sea level, Nature Climate Change, 11, 514–520, 2021

Domingues, R., Goni, G., Baringer, M., and Volkov, D.: What caused the accelerated sea level changes along the US East Coast during2010–2015?, Geophysical Research Letters, 45, 13–367, https://doi.org/10.1029/2018GL081183, 2018.

Ezer, T.: Regional Differences in Sea Level Rise Between the Mid-Atlantic Bight and the South Atlantic Bight: Is the Gulf Stream to Blame?,855Earth's Future, 7, 771–783, 2019.

Kuroda, H., Shimizu, M., and Setou, T.: Interannual variability of subsurface temperature in summer induced by the Kuroshio over BungoChannel, Tosa Bay, and Kii Channel, south of Japan, Continental Shelf Research, 30, 152–162, https://doi.org/10.1016/j.csr.2009.10.013,2010.

Sasaki, Y. N., Minobe, S., and Miura, Y.: Decadal sea-level variability along the coast of Japan in response to ocean circulation changes, Journal of Geophysical Research: Oceans, 119, 266–275, 2014.

---

## Author Response (AR2)

Dear Topic Editor,

Thanks for suggesting corrections.

**Comment**

"Thank you for the revised manuscript. All comments from both reviewers have been appropriately addressed. In the final reading, I have noticed 2 possible typos;

L.484 "Figure 4(a)" → "Figure 4(b)"
Figure 7 caption L.2, "red dashed" → "orange dashed"

Please fix them if they are typos. Thank you for choosing Ocean Science."

**Response & Changes in the manuscript**
These typos were fixed.

Samuel Diabaté,
On behalf of all co-authors.